# Virus-specific memory T cell responses unmasked by immune checkpoint blockade cause hepatitis

James A. Hutchinson [1✉], Katharina Kronenberg[1], Paloma Riquelme[1], Jürgen J. Wenzel [2], Gunther Glehr [3], Hannah-Lou Schilling [1], Florian Zeman[4], Katja Evert[5], Martin Schmiedel[6], Marion Mickler[7], Konstantin Drexler[7], Florian Bitterer[1], Laura Cordero[1], Lukas Beyer[8], Christian Bach[9], Josef Koestler[2], Ralph Burkhardt[10], Hans J. Schlitt [1], Dirk Hellwig[6], Jens M. Werner[1], Rainer Spang[3], Barbara Schmidt [2], Edward K. Geissler[1,11] & Sebastian Haferkamp[7]

Treatment of advanced melanoma with combined PD-1/CTLA-4 blockade commonly causes serious immune-mediated complications. Here, we identify a subset of patients predisposed to immune checkpoint blockade-related hepatitis who are distinguished by chronic expansion of effector memory CD4$^+$ T cells (T$_{EM}$ cells). Pre-therapy CD4$^+$ T$_{EM}$ cell expansion occurs primarily during autumn or winter in patients with metastatic disease and high cytomegalovirus (CMV)-specific serum antibody titres. These clinical features implicate metastasis-dependent, compartmentalised CMV reactivation as the cause of CD4$^+$ T$_{EM}$ expansion. Pre-therapy CD4$^+$ T$_{EM}$ expansion predicts hepatitis in CMV-seropositive patients, opening possibilities for avoidance or prevention. 3 of 4 patients with pre-treatment CD4$^+$ T$_{EM}$ expansion who received αPD-1 monotherapy instead of αPD-1/αCTLA-4 therapy remained hepatitis-free. 4 of 4 patients with baseline CD4$^+$ T$_{EM}$ expansion given prophylactic valganciclovir and αPD-1/αCTLA-4 therapy remained hepatitis-free. Our findings exemplify how pathogen exposure can shape clinical reactions after cancer therapy and how this insight leads to therapeutic innovations.

[1] Department of Surgery, University Hospital Regensburg, Regensburg, Germany. [2] Institute of Clinical Microbiology and Hygiene, University Hospital Regensburg, Regensburg, Germany. [3] Institute of Functional Genomics and Statistical Bioinformatics, University of Regensburg, Regensburg, Germany. [4] Center for Clinical Studies, University Hospital Regensburg, Regensburg, Germany. [5] Institute of Pathology, University Hospital Regensburg, Regensburg, Germany. [6] Department of Nuclear Medicine, University Hospital Regensburg, Regensburg, Germany. [7] Department of Dermatology, University Hospital Regensburg, Regensburg, Germany. [8] Institute of Radiology, University Hospital Regensburg, Regensburg, Germany. [9] Department of Medicine V, University Hospital Erlangen, Erlangen, Germany. [10] Institute of Clinical Chemistry and Laboratory Medicine, University Hospital Regensburg, Regensburg, Germany. [11] Personalised Tumour Therapy, Fraunhofer Institute for Experimental Medicine and Toxicology, Regensburg, Germany. ✉email: james.hutchinson@ukr.de

Treatment with anti-programmed cell death protein-1 (αPD-1) alone or combined with anti-cytotoxic T-lymphocyte-associated protein 4 (αCTLA-4) has greatly improved the prognosis of metastatic melanoma with a substantial fraction of patients achieving long-lasting partial or complete clinical responses[1–6]. The principal limitation of αPD-1/αCTLA-4 dual therapy is the relatively high rate of immune-related complications, which commonly include colitis, hepatitis and thyroiditis, but may also include pneumonitis, myocarditis and encephalitis[7,8]. Such adverse events range from mild reactions to life-threatening presentations, but are nevertheless clinically significant incidents that influence patient management[9,10]. Avoidance of complications has led to a somewhat complicated treatment decision-making process (Fig. 1a) that seeks to balance individual risk and capacity to tolerate adverse reactions against the superior clinical responses achieved with dual αPD-1/αCTLA-4 therapy over αPD-1 monotherapy[11–13].

Our aetiological understanding of colitis, hepatitis and thyroiditis caused by immune checkpoint blockade remains incomplete[14]; however, these treatment-related conditions are clearly different from known autoimmune diseases[15,16]. Such adverse reactions are usually attributed to generalised dysregulation of T cells[17,18] or autoantibodies[15]. Here, we explore a different view—namely, that individual patients are predisposed to particular treatment-related complications by virtue of pathogen-specific memory T cells present in their repertoire before immune checkpoint blockade is given.

Cytomegalovirus (CMV) seroprevalance amongst European adults is over 50%. Primary CMV infection usually presents during young adulthood as an acute febrile illness, after which life-long latency is established. The exact sites and mechanisms of CMV latency are not fully resolved, but myelomonocytic cells in blood and tissues are major reservoirs[19]. CMV reactivation characteristically follows immune compromise caused by immunosuppressive therapy or other viral infections[20]. Clinical manifestations of CMV reactivation are variable, ranging from classic febrile illness to tissue-invasive disease presenting as colitis, hepatitis, pneumonitis or encephalitis[21]. Reactivation of CMV causes cellular injury through the cytopathic effect of viral replication, but more conspicuously through triggering T cell immunity. Pneumonitis in transplant recipients illustrates how CMV-reactive T cells provoke extensive, life-threatening inflammation despite low-level viral replication[22]. Even in anatomical locations from which T cells are normally excluded, such as the retina, T cell-mediated responses significantly contribute to CMV-related disease. Thus, low-level and compartmentalised reactivation of CMV can be responsible for serious acute immunopathology in humans[23–25].

Cycles of CMV latency and reactivation lead to a now well-recognised phenomenon of memory inflation, which is typified by failure of primed CMV-specific CD8+ T cell populations to contract and give rise to central memory T cells[26]. Instead, a dominant pool of effector memory CD8+ T cells that does not become functionally exhausted is maintained[27]. Such inflationary CD8+ T cells achieve high frequencies over time, sustained through repetitive stimulation by viral reactivation or long-term persistence of viral antigens[28]. Memory inflation is described as an exclusive feature of CD8+ effector T cell responses, although durable expansion of CMV-specific CD4+ T cells can occur in transplant recipients[29].

This study identifies pre-therapy expansion of CD4+ effector memory T cells ($T_{EM}$) in CMV-infected patients with metastatic melanoma as a prognostic marker of αPD-1/αCTLA-4-related hepatitis. Opting for αPD-1 monotherapy or prophylactic valganciclovir in CMV IgG+ patients with unresectable metastatic disease and pre-treatment expansion of CD4+ $T_{EM}$ cells reduces

their risk of hepatitis. We conclude that sub-clinical reactivation of CMV plays a previously unsuspected role in the immuno-pathogenesis of αPD-1/αCTLA-4-related hepatitis in a subgroup of patients.

## Results

**Clinical features do not predict αPD-1/αCTLA-4-related hepatitis.** If immune-related complications of αPD-1/αCTLA-4 therapy resulted solely from treatment-induced immunological effects[30,31], we might expect hepatitis, colitis and thyroiditis to coincide often[32] (Fig. 1b). However, no such associations were found in a cohort of 89 patients with metastatic melanoma treated with αPD-1/αCTLA-4 dual therapy (Fig. 1c, Supplementary Fig. 1 and Supplementary Tables 1–3). This raises the possibility that some individuals are predisposed to particular adverse reactions.

In this study, we focused on factors that predispose to αPD-1/αCTLA-4-related hepatitis. Except for patients who developed hepatitis being generally younger (Supplementary Table 4) no associations were found between incidence of hepatitis and pre-treatment clinical variables (Supplementary Tables 4 and 5). Conventional biochemical markers of liver dysfunction, systemic inflammation or tumour burden did not discriminate between patients who did or did not develop hepatitis (Fig. 1d–j). We found no associations between incidence of hepatitis and seropositivity for hepatitis B virus (HBV), hepatitis C virus (HCV) or hepatitis E virus (HEV; Fig. 1k–m). Incidence of hepatitis was not greater with more rounds of αPD-1/αCTLA-4 therapy; in fact, as immune-related complications are reason to withhold treatment, patients who developed hepatitis tended to receive fewer rounds of αPD-1/αCTLA-4 (Fig. 1n). Autoimmune hepatitis-related autoantibodies were not associated with risk of hepatitis (Supplementary Table 6). In short, case history and routine clinical investigations did not predict hepatitis after dual therapy in our study cohort.

**CD4+ $T_{EM}$ cell expansion predicts αPD-1/αCTLA-4-related hepatitis.** We next profiled circulating leucocyte subsets immediately prior to start of dual therapy in a training set of $n = 44$ patients with unresectable metastatic melanoma (Supplementary Figs. 2–8). Comparing the frequency of 50 principal leucocyte subsets between patients who subsequently did ($n = 18$) or did not ($n = 26$) develop hepatitis revealed significant over-representation of CD4+ $T_{EM}$ cells associated with hepatitis (Fig. 2a, b). This observation was confirmed in a validation set ($n = 45$) of similar patients (Fig. 2c). Baseline expansion of CD4+ $T_{EM}$ cells was a fair discriminator (AUROC = 0.706) of patients who did or did not later develop hepatitis in the validation set (Fig. 2d). Strikingly, CD4+ $T_{EM}$ cell frequencies in all patients with completely resected disease were normally distributed with no extreme cases of baseline CD4+ $T_{EM}$ cell expansion, allowing us to set a threshold below which 99% of patients should fall (Fig. 2e). Applying this stringent cut-off to our validation cohort of patients with unresectable metastatic disease (Fig. 2f) allowed us to correctly predict hepatitis in 6 of 6 patients with CD4+ $T_{EM}^{\geq 21\%}$ (specificity = 100%) and correctly identify 6 of 20 patients who developed hepatitis (sensitivity = 30.0%). CD4+ $T_{EM}^{\geq 21\%}$ patients were not predisposed to colitis (Fig. 2g), thyroiditis or superior clinical responses to therapy (Supplementary Table 7). Hepatitis was not more severe in CD4+ $T_{EM}^{\geq 21\%}$ patients compared to CD4+ $T_{EM}^{<21\%}$ patients (Fig. 2h) and no difference in time-to-first presentation was observed (Fig. 2i). Hence, baseline CD4+ $T_{EM}$ expansion is a specific, but relatively insensitive prognostic marker of individuals who are predisposed to hepatitis after αPD-1/αCTLA-4 therapy. Owing to its low sensitivity, classifying patients as CD4+ $T_{EM}^{<21\%}$ has limited clinical utility; however, specifically identifying cases at high-risk

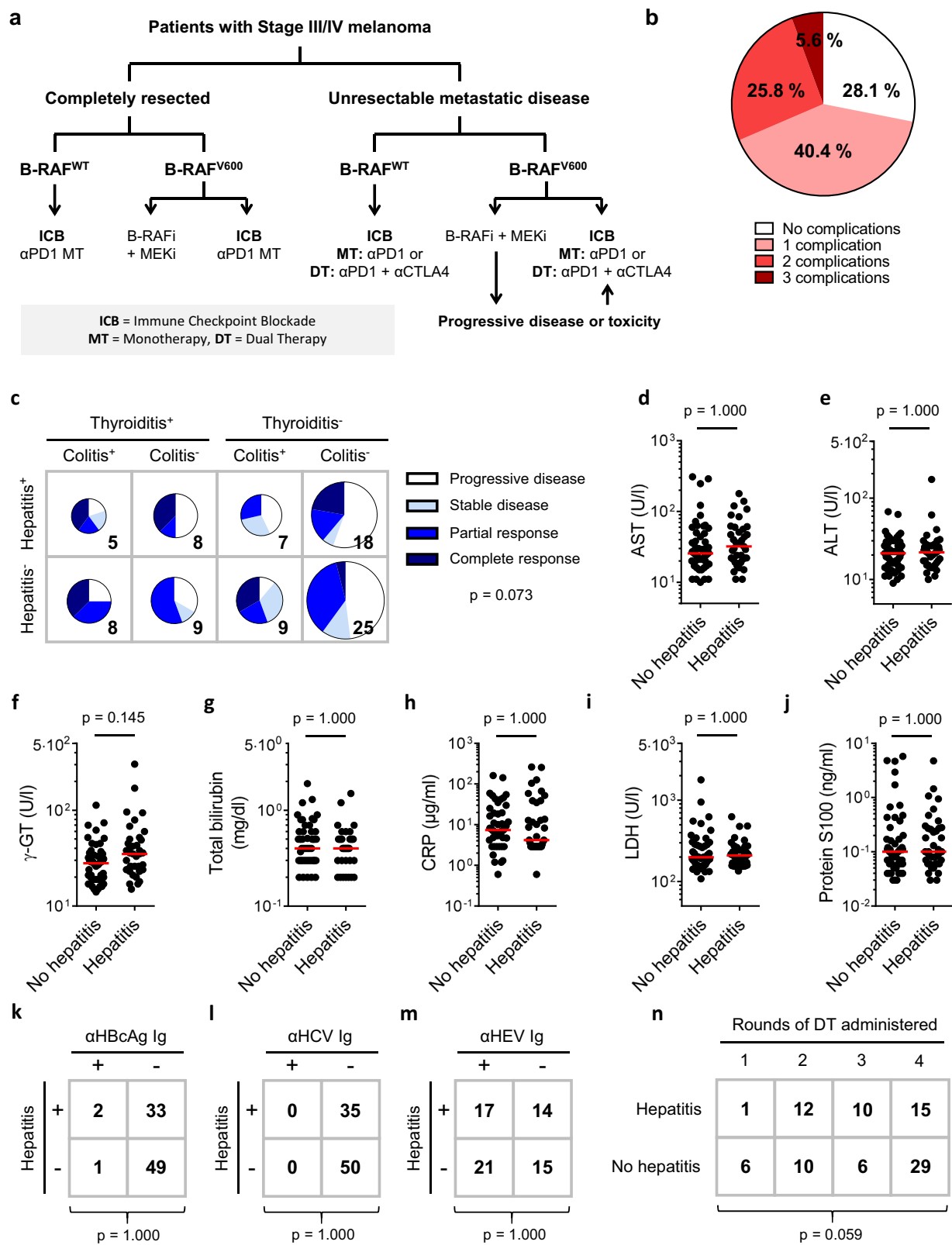

of hepatitis before treatment is useful from a research perspective because we can then prospectively study the immunological basis of their predisposition.

**Opting for αPD-1 monotherapy reduces risk of hepatitis.** Although αPD-1 monotherapy achieves fewer sustained clinical responses in patients with metastatic melanoma, it causes fewer adverse reactions than αPD-1/αCTLA-4 dual therapy[1]. Therefore, there is a clear rationale for avoiding αPD-1/αCTLA-4 dual therapy in patients at risk of hepatitis. Four $CD4^+$ $T_{EM}^{\geq21\%}$ patients with metastasic disease presented during our study, but were electively treated with αPD-1 monotherapy (Fig. 2j).

**Fig. 1 Individual predisposition to hepatitis after αPD-1/αCTLA-4 treatment. a** Individualised treatment of melanoma is guided by tumour staging, presence of B-RAF mutations and fitness-for-toxicity. **b** Colitis, hepatitis and thyroiditis are common immune-related complications of dual therapy with Nivolumab plus Ipilimumab; 31.4% of patients experienced two or more of these immune-related adverse reactions ($n = 89$). **c** Colitis, hepatitis and thyroiditis occurred independently and were not significantly associated with clinical response ($n = 89$; F.E). **d–h** Patients who developed hepatitis of any grade following dual therapy lacked biochemical signs of liver inflammation before treatment. In particular, no clinically meaningful differences in plasma levels of **d** aspartate transaminase (AST; $n = 87$; M.W.; Bonferroni-corrected $p$-value, $m = 5$), **e** alanine transaminase (ALT; $n = 89$; M.W.; Bonferroni-corrected $p$-value, $m = 5$), **f** gamma glutamyl transaminase (γ-GT; $n = 89$; M.W.; Bonferroni-corrected $p$-value, $m = 5$), **g** total bilirubin ($n = 87$; M.W.; Bonferroni-corrected $p$-value, $m = 5$), or **h** C-reactive protein (CRP; $n = 85$; M.W.; Bonferroni-corrected $p$-value, $m = 5$) were observed between patients who developed hepatitis and those who did not. Median values are indicated by a red line. **i, j** Biochemical markers of tumour burden were not different between patients who developed hepatitis and those who did not. **i** Pre-treatment levels of lactate dehydrogenase ($n = 89$; M.W.; Bonferroni-corrected $p$-value, $m = 4$). Median values are indicated by a red line. **j** Pre-treatment levels of protein S100 ($n = 89$; M.W.; Bonferroni-corrected $p$-value, $m = 4$). **k–m** No association was observed between seropositivity for **k** hepatitis B virus core antigen (HBcAg; $n = 85$; F.E.; Bonferroni-corrected $p$-value, $m = 3$), **l** hepatitis C virus (HCV; $n = 85$; F.E.; Bonferroni-corrected $p$-value, $m = 3$), or **m** hepatitis E virus (HEV; $n = 67$; F.E.; Bonferroni-corrected $p$-value, $m = 3$) and development of hepatitis following dual therapy. **n** No association was observed between rounds of αPD-1/αCTLA-4 administered and development of hepatitis ($n = 89$; M.W.).

Whereas 12 of 12 CD4$^+$ T$_{EM}$$^{≥21\%}$ patients treated with αPD-1/αCTLA-4 dual therapy developed hepatitis, 3 of 4 CD4$^+$ T$_{EM}$$^{≥21\%}$ patients who received αPD-1 monotherapy remained hepatitis free. Reducing the incidence hepatitis by opting for αPD-1 monotherapy hints that CTLA-4 blockade may be mechanistically important for development of hepatitis in CD4$^+$ T$_{EM}$$^{≥21\%}$ patients[33].

**CD4$^+$ T$_{EM}$ frequency is an independent predictor of hepatitis**. We next investigated possible causes of baseline CD4$^+$ T$_{EM}$ expansion. To establish CD4$^+$ T$_{EM}$ expansion as an independent risk factor, we first sought to exclude possible confounding covariables. Baseline CD4$^+$ T$_{EM}$$^{≥21\%}$ was not related to sex, age or body-mass index (Supplementary Table 7 and Fig. 3a, b). No significant correlation was observed between baseline CD4$^+$ T$_{EM}$ % and serum levels of C-reactive protein (CRP), aspartate aminotransferase (AST), alanine aminotransferase (ALT), gamma-glutamyl transferase (γ-GT) or total bilirubin (Fig. 3c–g). Because glucose metabolism provides a reliable measure of liver inflammation[34], we were able to further rule out liver inflammation prior to therapy as a predisposing cause of CD4$^+$ T$_{EM}$$^{≥21\%}$ hepatitis by estimating hepatic glucose uptake in patients who underwent $^{18}$F-fluoro-2-deoxy-D-glucose ($^{18}$F-FDG) PET/CT studies as part of routine tumour staging shortly before starting immunotherapy (Fig. 3h). This retrospective comparison showed $^{18}$F-FDG standard uptake ratios (SUR$_{mean}$ liver) were higher in CD4$^+$ T$_{EM}$$^{<21\%}$ patients than CD4$^+$ T$_{EM}$$^{≥21\%}$ patients with hepatitis (Fig. 3i). Therefore, our biochemical and radiological investigations argue against pre-treatment liver inflammation as a common cause for baseline CD4$^+$ T$_{EM}$ expansion and treatment-related hepatitis.

Because CD4$^+$ T$_{EM}$ expansion is principally a feature of metastatic melanoma (Fig. 2e), we next asked whether CD4$^+$ T$_{EM}$% was somehow associated with tumour burden. No relationship was found between CD4$^+$ T$_{EM}$% and the number of organs containing metastases, presence of hepatic metastases or serum lactate dehydrogenase (LDH) and protein-S100 serum levels (Fig. 3j–m and Supplementary Table 7). There was a tendency for CD4$^+$ T$_{EM}$$^{≥21\%}$ patients to have received previous therapy of any type (including BRAFi/MEKi or Talimogen laherparepvec) but besides implying a longer disease course before starting dual therapy, we see no mechanistic interpretation because these drugs share no common pharmacological action (Fig. 3n). In short, we found no obvious clinical association to explain the predisposition of CD4$^+$ T$_{EM}$$^{≥21\%}$ patients to hepatitis.

**CD4$^+$ T$_{EM}$ cell expansion is associated with chronic T cell activation**. Although CD4$^+$ T cell counts were not different between CD4$^+$ T$_{EM}$$^{<21\%}$ and CD4$^+$ T$_{EM}$$^{≥21\%}$ patients (Supplementary Table 7), there were significantly fewer circulating naïve T cells in CD4$^+$ T$_{EM}$$^{≥21\%}$ patients (Fig. 4a). This reduction was complemented by increased absolute numbers of CD4$^+$ T$_{EM}$ and T$_{EMRA}$ cells (Fig. 4b, c) whereas central memory T cell (T$_{CM}$) counts tended to be lower in CD4$^+$ T$_{EM}$$^{≥21\%}$ patients (Fig. 4d). We next defined other leucocyte subsets that correlated with CD4$^+$ T$_{EM}$ % in both our training and validation cohorts (Fig. 4e). Most covariant populations were T cells, particularly CD4$^+$ T cell subsets. Changes in the CD4$^+$ T cell compartment of T$_{EM}$$^{≥21\%}$ patients were consistent with chronic activation, particularly downregulation of CD27 and upregulation of CD57, HLA-DR and CD279 expression (Fig. 4f–i). Similar differences were detected in CD8$^+$ T cells (Supplementary Fig. 9). Notably, CD4$^+$ T$_{EM}$$^{≥21\%}$ patients tended to have higher frequencies of CD160$^+$ CD244$^+$ exhausted CD8$^+$ T cells than CD4$^+$ T$_{EM}$$^{<21\%}$ patients (Fig. 4j).

Most adults are persistently infected with Torque Teno Virus (TTV), a non-enveloped, circular, single-stranded DNA virus of the Anellovirus family that causes no apparent disease. T cell immunity controls TTV replication in healthy people, whereas TTV copy number in immunocompromised individuals is increased; therefore, TTV load in plasma is a useful marker of immune competence. Despite accumulating chronically activated and exhausted T cells, TTV load in CD4$^+$ T$_{EM}$$^{≥21\%}$ versus CD4$^+$ T$_{EM}$$^{<21\%}$ patients was not different, implying that CD4$^+$ T$_{EM}$$^{≥21\%}$ patients are not generally immunocompromised (Fig. 4k).

**CD4$^+$ T$_{EM}$ cell expansion correlates with immunity to CMV**. To briefly summarise, baseline expansion of CD4$^+$ T$_{EM}$ cells identifies a subset of patients with metastasic melanoma who are predisposed to αPD-1/αCTLA-4-related hepatitis. These patients show signs of persistent or recurrent CD4$^+$ and CD8$^+$ T cell responses[35], but are otherwise clinically and immunologically unremarkable. It is usually very difficult to isolate factors responsible for such a non-specific immunological picture[36]. Fortunately, a surprising pattern emerged from our dataset: CD4$^+$ T$_{EM}$$^{≥21\%}$ patients mostly presented between September and February (Fig. 5a). This seasonal influence was apparent over all 4 years of the study, leading us to think that CD4$^+$ T$_{EM}$ expansion might be driven by a pathogen. Active infections with hepatitis viruses, including HBV and HCV, are relative contraindications to αPD-1/αCTLA-4 therapy (Supplementary Table 7), so we were forced to consider possible involvement of other hepatotropic viruses, such as HEV and herpesgroup viruses.

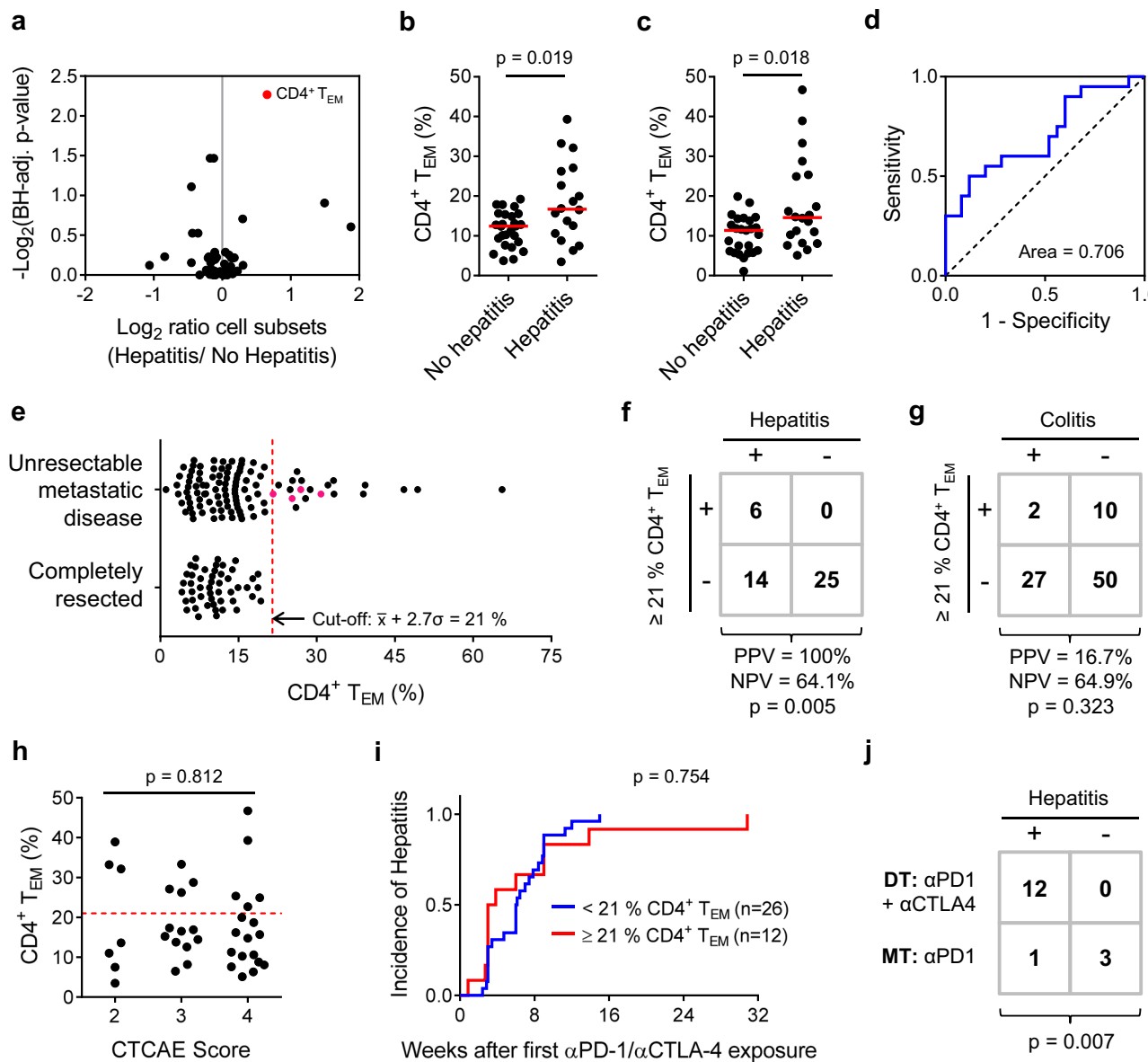

**Fig. 2 Circulating CD4$^+$ T effector memory cell frequency predicts hepatitis after Nivolumab plus Ipilimumab treatment. a** Peripheral blood samples were collected from melanoma patients with metastatic disease receiving $\alpha$PD-1/$\alpha$CTLA-4 therapy immediately before administration of the first dose ($n = 89$). Leucocyte subsets differentially represented in patients with or without hepatitis were identified in a randomly assigned training set (B.H.-corrected t-tests; $n = 44$; $m = 50$; FDR = 0.25). Red dots indicate significantly differently represented subsets. Example gating strategies for analysis of flow cytometry data are provided as Supplementary Figs. 2–8. **b** CD4$^+$ T$_{EM}$ % in training set patients with or without treatment-related hepatitis ($n = 44$; M.W.). Median values are indicated by a red line. **c** CD4$^+$ T$_{EM}$ % in validation set patients with or without treatment-related hepatitis ($n = 45$; M.W.). **d** ROC analysis of CD4$^+$ T$_{EM}$ % as a discriminatory marker for treatment-related hepatitis in the validation set ($n = 45$). **e** Comparison of the bimodal distribution of CD4$^+$ T$_{EM}$ % in patients with unresectable metastatic disease ($n = 107$) and the normal distribution ($n = 49$; K2 = 2.79; $p = 0.248$) of CD4$^+$ T$_{EM}$ % in patients with completely resected tumours. A cut-off of CD4$^+$ T$_{EM} \geq 21\%$ was set (indicated by a dashed red line) below which 99% of completely resected tumour cases should fall. Four pink points represent CD4$^+$ T$_{EM}^{\geq 21\%}$ patients with metastatic disease who were electively treated with $\alpha$PD-1 monotherapy. **f** In the validation set, 68.9% patients were correctly classified using a cut-off of CD4$^+$ T$_{EM} \geq 21$ %, whereas 55.6% were correctly classified under the no-information model ($n = 45$; F.E.). **g** CD4$^+$ T$_{EM} \geq 21$ % is not a marker of predisposition to $\alpha$PD-1/$\alpha$CTLA-4-related colitis ($n = 89$; F.E.). **h** CD4$^+$ T$_{EM}^{\geq 21\%}$ patients did not experience more severe hepatitis than CD4$^+$ T$_{EM}^{<21\%}$ patients ($n = 38$; F.E.). Dashed red line indicates a cut-off of CD4$^+$ T$_{EM} \geq 21\%$. **i** Time-to-first presentation of hepatitis was not different between CD4$^+$ T$_{EM}^{\geq 21\%}$ ($n = 12$) and T$_{EM}^{<21\%}$ ($n = 26$) patients (log-rank). **j** Twelve of 12 patients with unresectable metastatic melanoma and CD4$^+$ T$_{EM} \geq 21\%$ developed hepatitis after $\alpha$PD-1/$\alpha$CTLA-4 dual therapy. By contrast, 3 of 4 CD4$^+$ T$_{EM}^{\geq 21\%}$ patients treated with $\alpha$PD-1 monotherapy did not develop hepatitis (F.E.; $p = 0.007$).

Latent CMV infection is prevalent amongst European adults and involves many organs, including liver. Compartmentalised reactivation of CMV infection occurs in immunocompromised individuals and those with concomitant viral upper respiratory tract infections[20]. In our patients, we found a strong association between serum CMV-specific IgG antibody levels and CD4$^+$ T$_{EM}$ status (Fig. 5b). Furthermore, T cell reactivity against CMV pp65 peptides in an IFN-$\gamma$ enzyme-linked immunospot (ELISPOT) assay was tightly associated with CD4$^+$ T$_{EM}$ status (Fig. 5c). These results led us to hypothesise a causal relationship between compartmentalised, sub-clinical CMV reactivation in liver and chronic enrichment of CD4$^+$ T$_{EM}$ cells in blood.

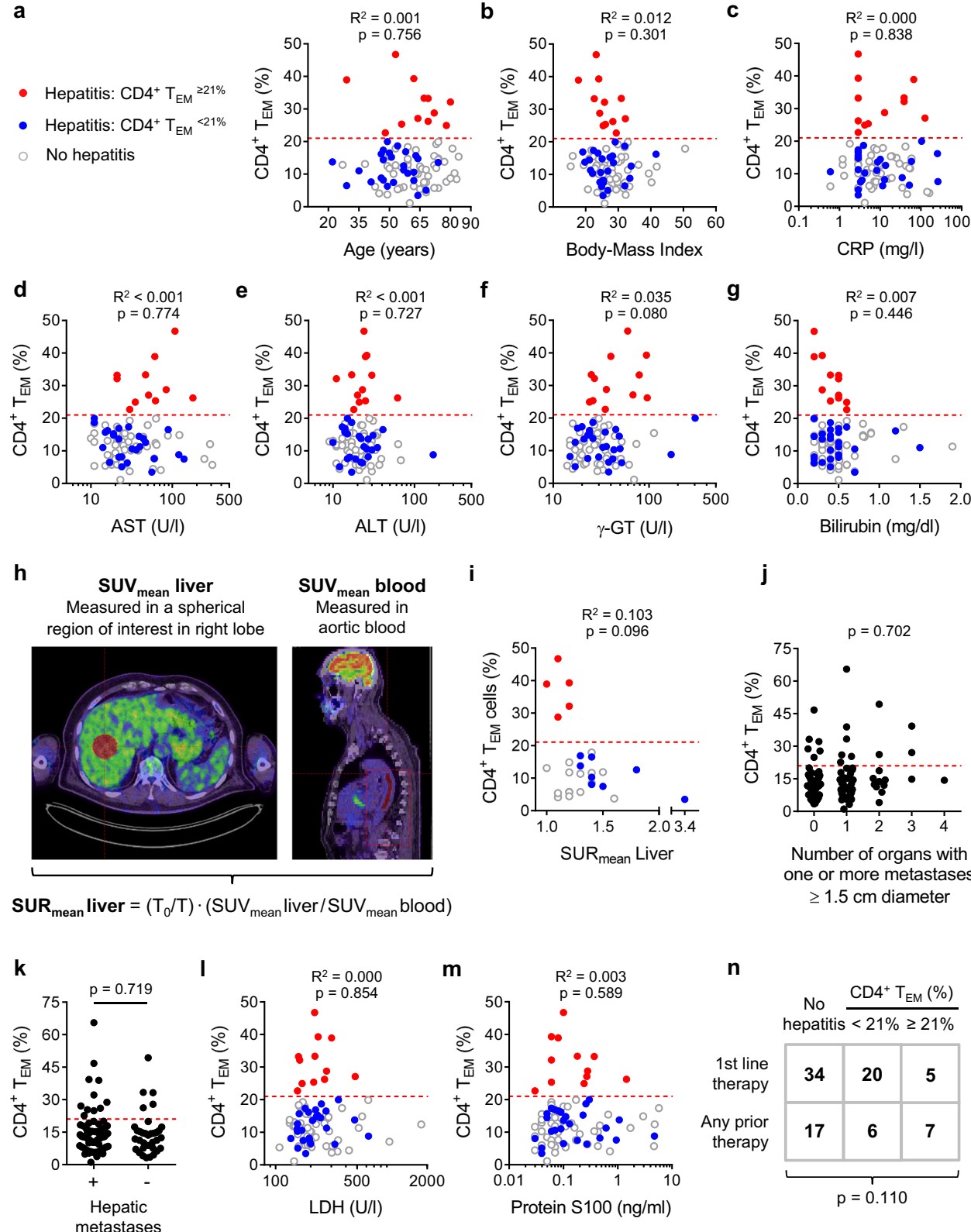

$$\textbf{SUR}_{\textbf{mean}} \textbf{ liver} = (T_0/T) \cdot (SUV_{mean} \text{ liver} / SUV_{mean} \text{ blood})$$

Because baseline CD4$^+$ T$_{EM}$ cell expansion was mainly a feature of CMV IgG$^+$ patients (Fig. 5d), we next asked whether incorporating CMV IgG status could improve our predictive model. When considering only CMV IgG$^+$ cases, CD4$^+$ T$_{EM}$ % was a more discriminatory marker of hepatitis (Fig. 5e).

Therefore, using our training set filtered for CMV IgG$^+$ cases, we revised our cut-off to CD4$^+$ T$_{EM} \geq 16\%$ in CMV IgG$^+$ cases in order to achieve a better compromise between specificity and sensitivity (Supplementary Fig. 10). The new cut-off performed well in the validation set with a correct classification rate of 81.0%

**Fig. 3 CD4$^+$ T$_{EM}$ frequency is an independent predictor of αPD-1/αCTLA-4-related hepatitis. a** CD4$^+$ T$_{EM}$ % did not correlate with age ($n = 89$; Pearson). Dashed red line indicates a cut-off of CD4$^+$ T$_{EM} \geq 21$%. **b** CD4$^+$ T$_{EM}$ % did not correlate with body mass index ($n = 89$; Pearson). **c** CD4$^+$ T$_{EM}$ % did not correlate with C-reactive protein (CRP) levels ($n = 85$; Pearson). **d** CD4$^+$ T$_{EM}$ % did not correlate with aspartate aminotransferase (AST) levels ($n = 87$; Pearson). **e** CD4$^+$ T$_{EM}$ % did not correlate with alanine aminotransferase (AST) levels ($n = 89$; Pearson). **f** CD4$^+$ T$_{EM}$ % did not correlate with gamma-glutamyltransferase (γ-GT) levels ($n = 89$; Pearson). **g** CD4$^+$ T$_{EM}$ % did not correlate with total bilirubin levels ($n = 89$; Pearson). **h** Glucose metabolism measured by $^{18}$F-fluoro-2-deoxy-D-glucose ($^{18}$F-FDG) uptake in PET/CT studies is a quantitative marker of liver inflammation. Calculating standard uptake ratio (SUR$_{mean}$ Liver) allows accurate quantification of $^{18}$F-FDG uptake in liver by correcting for differential clearance of tracer from blood and liver parenchyma. Higher SUR$_{mean}$ Liver values indicate more severe inflammation. $T_0$ = reference time point at 75 min post-injection and $T$ = actual scan time. **i** SUR$_{mean}$ Liver was lower in CD4$^+$ T$_{EM}^{\geq 21\%}$ patients than CD4$^+$ T$_{EM}^{<21\%}$ patients who developed hepatitis indicating less inflammation at baseline in the CD4$^+$ T$_{EM}^{\geq 21\%}$ subgroup ($n = 28$; Pearson). **j** CD4$^+$ T$_{EM}^{\geq 21\%}$ was not associated with the number of organs containing one or more metastases $\geq 1.5$ cm in diameter ($n = 103$; F.E.). **k** CD4$^+$ T$_{EM}^{\geq 21\%}$ was not associated with the presence of hepatic metastases ($n = 103$; F.E.). **l** CD4$^+$ T$_{EM}$ % did not correlate with lactate dehydrogenase (LDH) levels ($n = 88$; Pearson). **m** CD4$^+$ T$_{EM}$ % did not correlate with protein S100 levels ($n = 88$; Pearson). **n** CD4$^+$ T$_{EM}^{\geq 21\%}$ was not associated with any prior therapy ($n = 89$; F.E.).

(positive predictive value (PPV) = 85.7% and negative predictive value (NPV) = 78.6%) compared to 42.9% under the no-information model.

For descriptive purposes, we can consider the distribution of patients from the combined training and validation sets according to CMV IgG and CD4$^+$ T$_{EM}$ status (Fig. 5f). Our predictive model explains the occurrence of hepatitis in 34 of 89 (38.2%) of all patients (red boxes). Two of 17 (12%) CMV IgG$^+$ CD4$^+$ T$_{EM}^{\geq 16\%}$ patients (green box) did not develop hepatitis: one of these received four rounds of αPD-1/αCTLA-4 therapy and registered no complications; however, the other received only 1 round of αPD-1/αCTLA-4 before treatment with prednisolone from weeks 2 to 7 for colitis. We speculate 4 of 23 CMV IgG$^+$ CD4$^+$ T$_{EM}^{<16\%}$ patients who were incorrectly predicted as hepatitis-negative (pink box) were sampled at an early stage of CD4$^+$ T$_{EM}$ cell expansion and our test was not sensitive enough to discriminate the change. Our model does not explain why 19 of 49 CMV IgG$^-$ patients developed hepatitis (blue box) but we believe these represent an aetiologically distinct subset.

**CMV-reactive CD4$^+$ T cells are enriched in CD4$^+$ T$_{EM}^{high}$ patients.** We next asked whether CMV-reactive CD4$^+$ T cells were enriched in patients with baseline CD4$^+$ T$_{EM}$ expansion. Peripheral blood mononuclear cells (PBMC) were stimulated in culture for 18 h with CMV lysates before IFN-γ, TNFα, IL-17, IL-4 and CD69 expression was measured by flow cytometry (Fig. 6a and Supplementary Figs. 11 and 12). To detect hyporesponsive CMV-specific T cells, paired cultures were also stimulated in the presence of neutralising antibodies against PD-1 and CTLA-4. In contrast to CMV IgG$^-$ and CMV IgG$^+$ CD4$^+$ T$_{EM}^{<16\%}$ patients, we readily detected IFN-γ-producing, CMV-reactive CD4$^+$ T cells in CMV IgG$^+$ CD4$^+$ T$_{EM}^{\geq 16\%}$ patients (Fig. 6b). This difference implies a higher number of circulating IFN-γ-producing, CMV-reactive CD4$^+$ T cells (Fig. 6c). Proportions of CMV-reactive CD4$^+$ T cells expressing IL-17, IL-4 or only TNFα were too low to quantify accurately.

**Valganciclovir reverses αPD-1/αCTLA-4-related hepatitis.** Strong associations between T cell immunity against CMV, expansion of CD4$^+$ T$_{EM}$ and risk of treatment-related hepatitis imply an underlying pathogenic mechanism; however, to demonstrate a causal relationship, we needed to establish that inhibiting CMV replication was effective in treating or preventing hepatitis. Valganciclovir is an anti-viral agent licensed for therapy and prophylaxis of CMV infections in immunocompromised patients. It is a widely used and safe drug with a well-known and broadly acceptable profile of adverse effects. Therefore, we next gave valganciclovir treatment to two patients with αPD-1/αCTLA-4-related hepatitis.

The first patient was a comparatively fit 54-year-old man with a new diagnosis of metastatic melanoma (Fig. 7a). Staging CT revealed multiple lung and bone metastases, but no metastasis to liver or lymph nodes. At baseline, the patient was CMV-IgG$^+$ and CD4$^+$ T$_{EM}$ = 18.7% (Fig. 7b). Nivolumab plus Ipilimumab were given at 0, 3 and 6 weeks. A fourth round of αPD-1/αCTLA-4 at week 9 was omitted owing to joint pain and the patient received two rounds of Denosumab (αRANKL). Nivolumab monotherapy was administered in weeks 12, 16, 20 and 24. Staging CT at week 14 showed a partial clinical response with significant regression of pulmonary metastases; hence, checkpoint blockade was of clear clinical benefit.

The patient received a dose of αPD-1 monotherapy in week 32. Four weeks later, he presented in clinic with grade 4 hepatitis. Strong reactions against pp65 and IE-1 peptides were measured by ELISPOT. Accordingly, the patient was treated with 900 mg/day valganciclovir plus prednisolone. Liver function tests (LFTs) returned to normal, so over the subsequent 2 weeks, prednisolone was weaned and then valganciclovir was withdrawn. Seven days later, the patient returned to clinic with deranged LFTs. Valganciclovir and prednisolone were both reinstated, leading to a rapid normalisation of LFTs. Two days later, a for-cause liver biopsy was performed (Fig. 7c–e). The tissue was essentially normal, although a few dying hepatocytes and isolated lymphocytes were observed in the portal areas. Immunohistochemical staining for CMV and nested CMV PCR from fixed biopsy material were negative. The patient remained clinically stable for 9 days, so he was again weaned from prednisolone by the start of week 43; however, on this occasion, 900 mg/day valganciclovir treatment was continued. After complete withdrawal of prednisolone, while still receiving valganciclovir, the patient's LFTs remained stable for a further 4 weeks. At the start of week 47, valganciclovir was withdrawn for a second time. Subsequently, the patient did not develop hepatitis and required no further steroids. The patient received further doses of αPD-1 monotherapy in weeks 54 and 58 without signs of recurrent liver injury. To us, this suggests the T cell antigens driving hepatitis in this patient were absent after valganciclovir treatment. In our clinical opinion, despite the absence of detectable virus, the patient's fast recovery and unusual relapse of hepatitis are best explained by introduction, withdrawal and reintroduction of valganciclovir.

The second patient was a 49-year-old man who had an uncomplicated first year after αPD-1/αCTLA-4 and had been receiving Encorafenib/Binimetinib therapy for 7 weeks when he developed hepatitis in week 67 (Fig. 8a, b). Hepatitis failed to resolve despite withdrawal of B-RAFi/MEKi and escalating doses of prednisolone. 900 mg/day valganciclovir was introduced at the start of week 72 and a for-cause biopsy was performed on the same day (Fig. 8b). Four days later, the patient started a course of 1 mg/day mycophenolate mofetil. Strikingly, AST, ALT and γ-GT

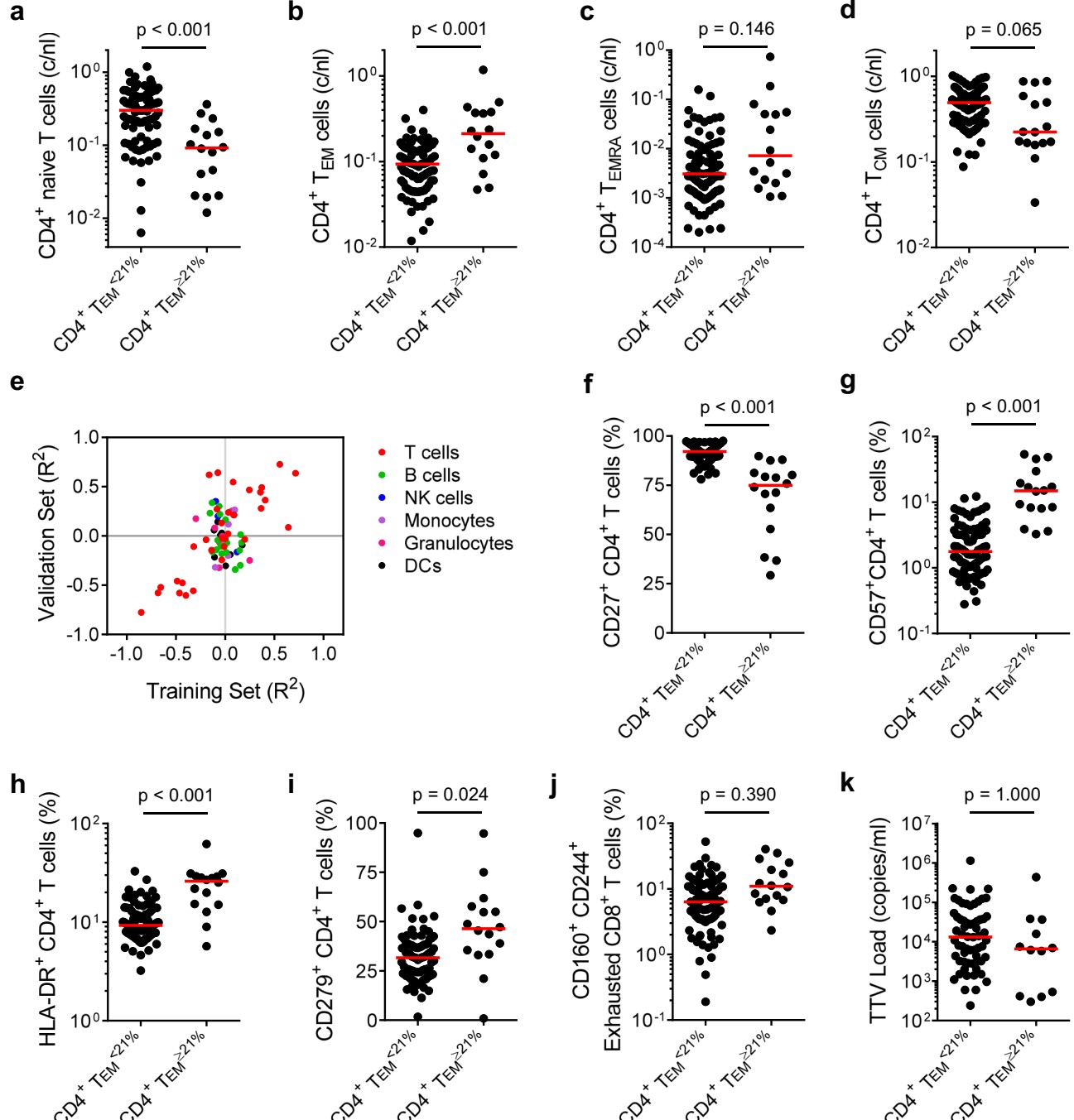

**Fig. 4 CD4$^+$ T$_{EM}$ cell enrichment is associated with chronic activation and increased numbers of effector memory CD4$^+$ T cells. a–d** Increased CD4$^+$ T$_{EM}$ frequency before treatment reflects a decrease in circulating naïve CD4$^+$ T cells and an increase in circulating numbers of CD4$^+$ T$_{EM}$ cells. **a** Baseline naïve CD4$^+$ T cell counts ($n = 103$; M.W.; Bonferroni correction: $m = 4$, $p = 5.2 \times 10^{-5}$). **b** Baseline CD4$^+$ T$_{EM}$ cell counts ($n = 103$; M.W.; Bonferroni correction: $m = 4$, $p = 8.0 \times 10^{-4}$). **c** Baseline CD4$^+$ T$_{EMRA}$ cell counts ($n = 103$; M.W.; Bonferroni-corrected $p$-value, $m = 4$). **d** Baseline CD4$^+$ T$_{CM}$ cell counts ($n = 103$; M.W.; Bonferroni-corrected $p$-value, $m = 4$). Median values are indicated by a red line. **e** Pairwise correlations between CD4$^+$ T$_{EM}$ cell frequency and other leucocyte subset frequencies in the training and validation sets ($n = 103$; Pearson). Example gating strategies for analysis of flow cytometry data are provided as Supplementary Figs. 2–8. **f** Baseline CD27$^+$ CD4$^+$ T cell frequencies ($n = 103$; M.W.; Bonferroni correction: $m = 60$, $p = 4.2 \times 10^{-10}$). **g** Baseline CD57$^+$ CD4$^+$ T cell frequencies ($n = 103$; M.W.; Bonferroni correction: $m = 60$, $p = 2.0 \times 10^{-9}$). **h** Baseline HLA-DR$^+$ CD4$^+$ T cell frequencies ($n = 103$; M.W.; Bonferroni correction: $m = 60$, $p = 4.1 \times 10^{-5}$). **i** Baseline CD279$^+$ CD4$^+$ T cell frequencies ($n = 103$; M.W.; Bonferroni-corrected $p$-value, $m = 60$). **j** Baseline CD160$^+$ CD244$^+$ CD8$^+$ T cell frequencies ($n = 103$; M.W.; Bonferroni-corrected $p$-value, $m = 60$). **k** Baseline Torque Teno Virus (TTV) loads. Uninfected patients were censored from analysis ($n = 74$; M.W.; Bonferroni-corrected $p$-value, $m = 60$). Median values are indicated by a red line.

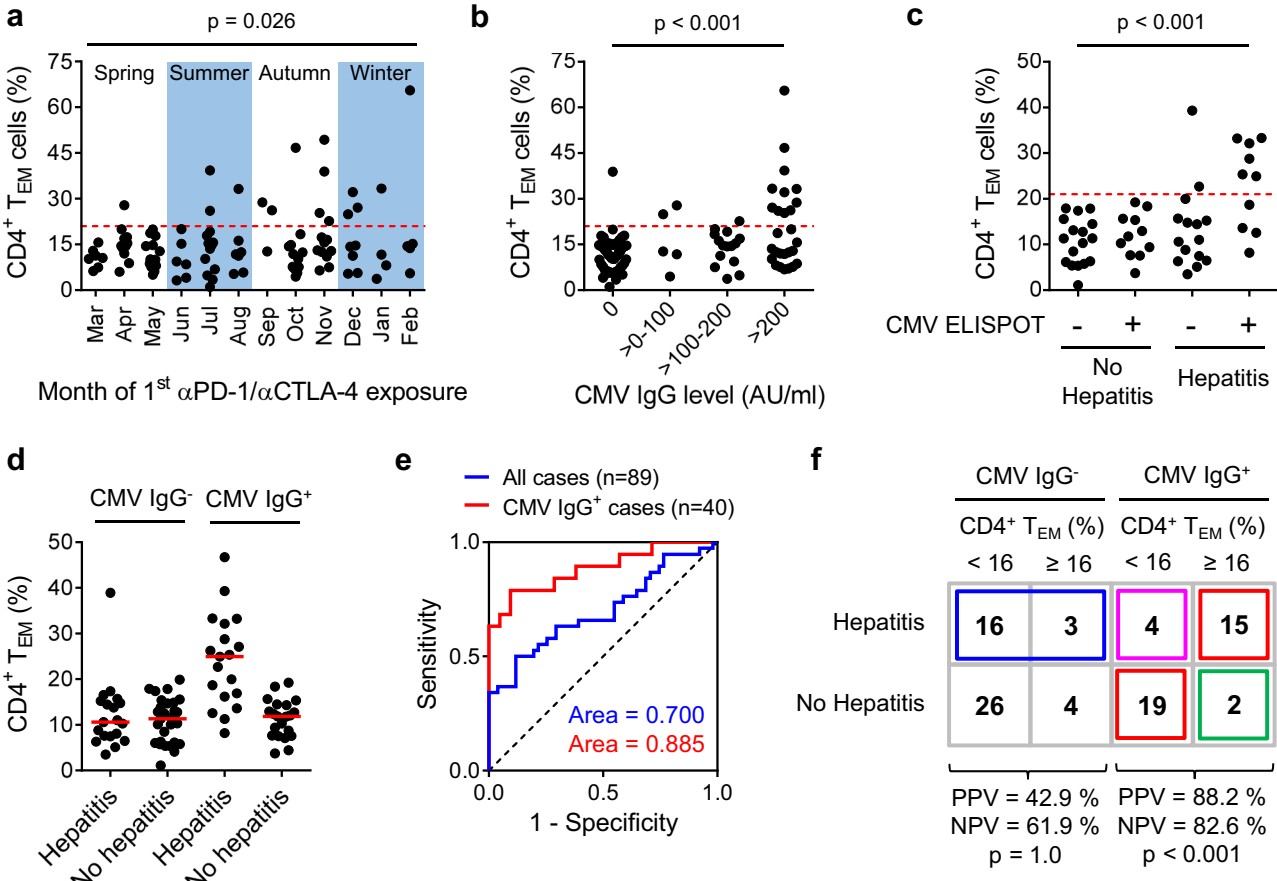

**Fig. 5 CD4+ T_EM cell expansion correlates with immunity to cytomegalovirus. a** Seasonal presentation of CD4+ T_EM≥21% patients between 2017 and 2020 (*n* = 103; F.E.). Dashed red line indicates a cut-off of CD4+ T_EM ≥ 21%. **b** CD4+ T_EM≥21% status was associated with high serum levels of anti-CMV IgG antibodies (*n* = 100; F.E.; *p* = 1.2 × 10−5). Dashed red line indicates a cut-off of CD4+ T_EM ≥ 21%. **c** CD4+ T_EM≥21% status was associated with CMV-reactivity in pp65 ELISPOT (*n* = 53; F.E.; *p* = 9.0 × 10−5). Dashed red line indicates a cut-off of CD4+ T_EM ≥ 21%. **d** Development of hepatitis was associated with CMV-seropositivity and CD4+ T_EM≥21% status (*n* = 89). Median values are indicated by a red line. **e** ROC analysis showing CD4+ T_EM % is a superior discriminator of patients at risk of hepatitis when considering only CMV IgG+ cases as opposed to all cases. **f** Classification of patients with unresectable metastatic melanoma who did or did not develop αPD-1/αCTLA-4-related hepatitis according to CMV IgG status and baseline CD4+ T_EM cell frequency using a revised cut-off of CD4+ T_EM ≥ 16%. Red boxes indicate 34 of 40 (85%) cases correctly classified by our model (*n* = 40; F.E.; *p* = 1.4 × 10−5). Green box indicates 2 of 17 (11.7%) cases predicted to develop hepatitis who did not. Pink box indicates 4 of 23 (17.4%) cases predicted not to develop hepatitis who did. Blue box indicates 19 of 49 CMV IgG− patients not considered by our model who developed hepatitis.

levels declined consistently from the day after valganciclovir was given, although total bilirubin continued to rise. Starting valganciclovir treatment appeared to be temporally related to increased reactivity in CMV IE-1 ELISPOT and a small decline in CD4+ T_EM frequency.

Liver biopsy revealed dense infiltration by lymphocytes, eosinophils and neutrophils with an accompanying ductal reaction (Fig. 8c–e). Extensive centrilobular hepatocyte necrosis was observed, which was also associated with inflammatory infiltration. These histopathological signs of significant liver damage could have been consistent with autoimmunity or viral infection. Both histological staining for CMV and CMV PCR from fixed biopsy material were negative. Although we were unable to detect virus, it is our clinical impression that markers of liver damage responded to valganciclovir treatment.

**Valganciclovir prophylaxis prevents αPD-1/αCTLA-4-related hepatitis.** We next asked whether valganciclovir prophylaxis could prevent αPD-1/αCTLA-4-related hepatitis in predisposed individuals. The 50th, 75th and 90th percentiles of time-to-hepatitis in CMV-IgG+ patients were 24, 52 and 84 days, respectively; therefore, we treated four CMV IgG+ CD4+

T_EM≥16% patients receiving αPD-1/αCTLA-4 therapy with prophylactic valganciclovir for between 4 and 23 weeks (Fig. 9a). Two of these patients registered mild, transient transaminitis that resolved without treatment, so were not diagnosed with hepatitis. Hence, we observed no hepatitis in the four valganciclovir-treated patients, whereas 15 of 17 (88.2%) CMV IgG+ CD4+ T_EM≥16% patients treated with αPD-1/αCTLA-4 and no valganciclovir prophylaxis developed clinical hepatitis (Fig. 9b). To us, these four cases are consistent with the idea that CMV plays an important aetiological role in development of hepatitis after αPD-1/αCTLA-4 therapy in a specific subset of patients.

**Discussion**

Our study seeks to explain why a subset of patients with advanced melanoma is predisposed to hepatitis following immune checkpoint blockade. In our model, metastatic disease promotes compartmentalised CMV reactivation in latently infected individuals, possibly triggered by an unknown winter virus infection; in turn, CMV stimulates expansion of CD4+ T_EM cells prior to immunotherapy. When αPD-1/αCTLA-4 therapy is given, these T_EM cells might inflict liver injury directly or promote bystander T cell responses. We propose CMV-driven T cell responses are a single,

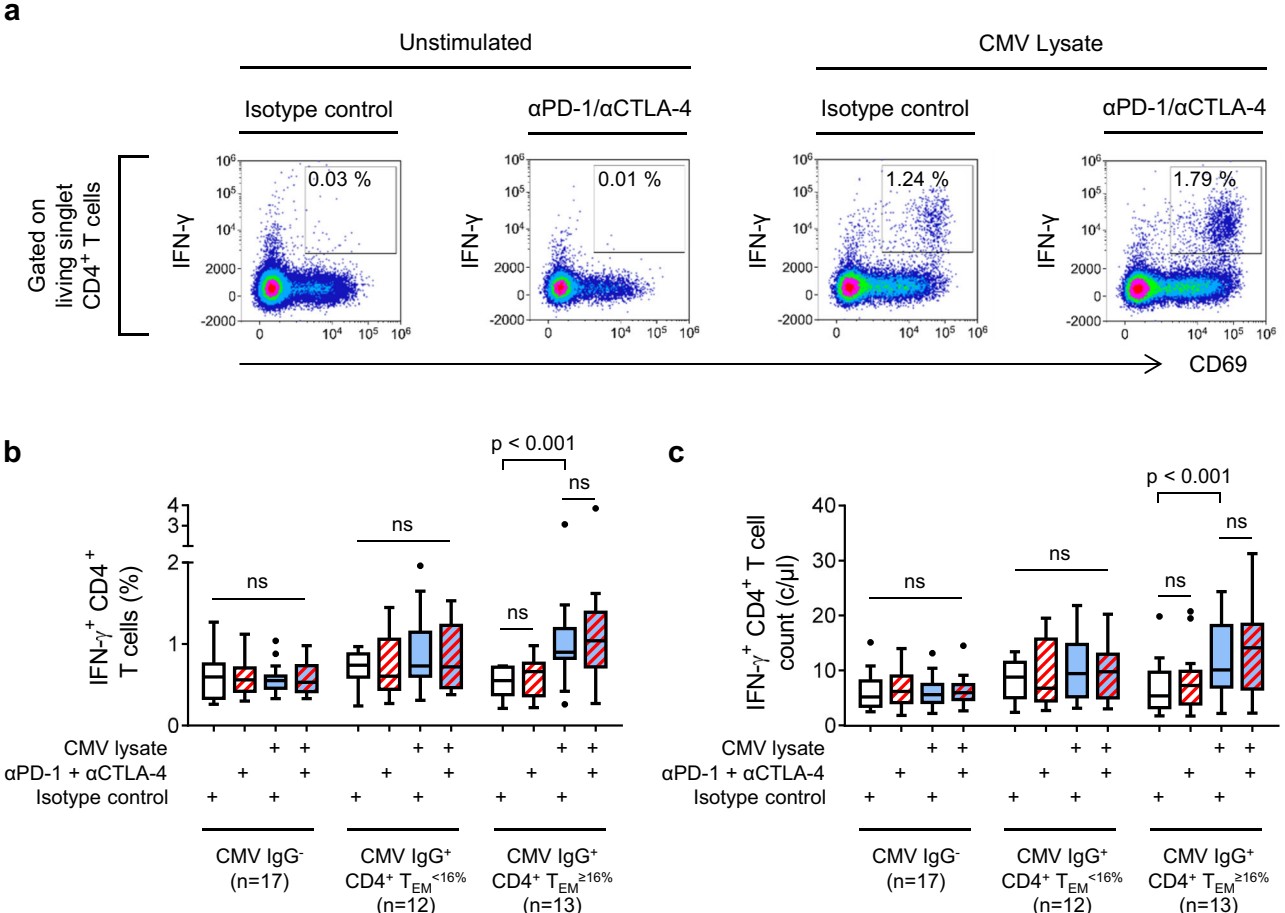

**Fig. 6 CMV-reactive CD4$^+$ T cells are enriched in patients with CD4$^+$ T$_{EM}$ cell expansion.** CMV-reactive CD4$^+$ T cells in patients with unresectable metastatic melanoma were assayed by in vitro stimulation with CMV lysates (blue boxes). Neutralising antibodies (red hatching) were used to detect T cells unable to respond to CMV antigens owing to expression of PD-1 or CTLA-4. T cell responses were quantified by flow cytometry analysis of cytokine expression. **a** Example data from a CMV IgG$^+$ CD4$^+$ T$_{EM}^{\geq16\%}$ patient who developed hepatitis. Gating of live, singlet, IFN-γ-producing CD4$^+$ T cells is illustrated in Supplementary Fig. 11. **b** Frequencies of IFN-γ-producing CD4$^+$ T cells in patients categorised according to CMV IgG serostatus and baseline CD4$^+$ T$_{EM}$ frequency ($n = 42$; two-way ANOVA with Tukey correction for multiple comparisons; n.s. = not significant). Boxplots represent the median, 25th and 75th percentiles, and Tukey whiskers. **c** Absolute numbers of circulating IFN-γ-producing CD4$^+$ T cells patients categorised according to CMV IgG serostatus and baseline CD4$^+$ T$_{EM}$ frequency ($n = 42$; two-way ANOVA with Tukey correction for multiple comparisons; n.s. = not significant). Boxplots represent the median, 25th and 75th percentiles, and Tukey whiskers.

but salient example of a general mechanism responsible for hepatitis after checkpoint blockade because other viruses or toxic agents could also drive chronic CD4$^+$ T cell activation before therapy. We speculate anti-viral immunity might also explain predisposition to other adverse reactions following αPD-1/αCTLA-4 therapy, such as colitis or pneumonitis[37]. The remarkable overlap between organs affected by tissue-invasive CMV disease and those susceptible to immune-related adverse reactions has not escaped our attention[38,39].

Being able to identify patients predisposed to hepatitis opens various possibilities for avoiding or preventing treatment-related hepatitis[40]. Three of four CD4$^+$ T$_{EM}^{\geq21\%}$ patients assigned to αPD-1 monotherapy instead of αPD-1/αCTLA-4-treatment did not develop hepatitis as predicted; therefore, baseline CD4$^+$ T$_{EM}$ expansion could be taken as a relative contraindication to dual therapy. However, a more promising strategy is to suppress occult CMV reactivation. In our study, valganciclovir prophylaxis apparently prevented hepatitis in four of four CMV IgG$^+$ CD4$^+$ T$_{EM}^{\geq16\%}$ patients. This approach must be refined in terms of dosing and duration of anti-viral therapy in randomised, controlled clinical trials, but could potentially achieve the superior

outcomes of αPD-1/αCTLA-4 dual therapy over αPD-1 monotherapy with fewer complications.

Because baseline CD4$^+$ T$_{EM}$ expansion was observed in patients with metastatic disease, but not in patients with fully resected melanoma, it is possible that disseminated melanoma promotes low-level CMV reactivation, perhaps through compromised T cell immunity, which then stimulates virus-specific CD4$^+$ T cell expansion. The prominent association between CD4$^+$ T$_{EM}$ expansion and CMV (in contrast to other herpesgroup virus infections) might simply reflect differences in tissue tropism and, we speculate, the capacity of compartmentalised CMV reactivation in liver to drive especially strong T cell responses. Seasonal variation in the incidence of CD4$^+$ T$_{EM}^{high}$ cases suggests that environmental factors influence expansion and contraction of the CD4$^+$ T$_{EM}$ cell pool[41,42]. In particular, we suspect common winter viruses somehow promote CMV reactivation in liver.

Chronic, low-level exposure to viral antigens preferentially expands T$_{EM}$ over T$_{CM}$[43–49], so CMV-reactive CD4$^+$ T cell expansion might be sustained by very little CMV replication. Ongoing T cell reactions against CMV in CD4$^+$ T$_{EM}^{high}$ patients are

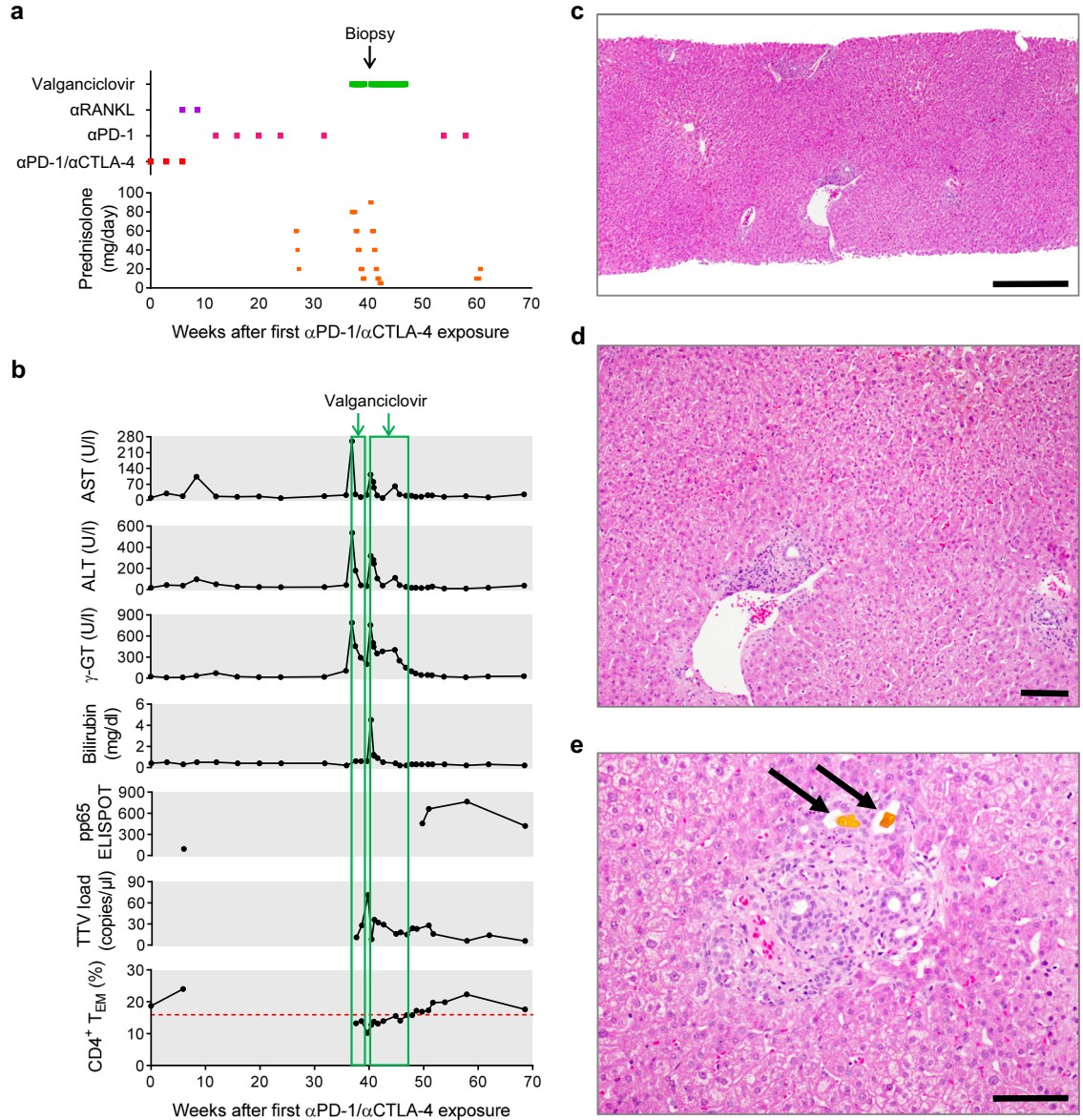

**Fig. 7 Case 1: valganciclovir treatment of late-onset αPD-1/αCTLA-4-related hepatitis.** A 54-year-old male patient who received αPD-1 (Nivolumab) plus αCTLA-4 (Ipilimumab) dual therapy for metastatic melanoma presented with late-onset hepatitis. **a** Course of treatment. **b** Change in hepatitis-related parameters over time and their association with introduction, withdrawal and re-introduction of valganciclovir treatment. **c** The patient presented at the end of week 40 with recrudescent hepatitis and was treated with 900 mg/day valganciclovir for 2 days prior to liver biopsy. Histopathological image of the liver biopsy (H&E staining; scale bar 500 μm). **d** Generally, the liver parenchyma appeared normal (H&E staining; scale bar 100 μm). **e** Only a few lymphocytes and sparse necrotic hepatocytes were observed in portal areas with small bile plugs (arrows) with no signs of hepatitis (H&E staining; scale bar 100 μm).

likely to suppress CMV replication almost completely. Our inability to detect CMV in baseline serum samples from CD4+ $T_{EM}^{high}$ patients or liver biopsies from patients with high-grade hepatitis is a challenge for our theory; however, it may simply be beyond our technical capability to detect very low levels of compartmentalised viral reactivation. In our view, the strong correlation between CMV-specific immunity and development of hepatitis, as well as the apparent effectiveness of valganciclovir, suggests a causal role for CMV in the aetiology of αPD-1/αCTLA-4-related hepatitis.

We do not presently know whether CMV-specific CD4+ $T_{EM}$ cells inflict liver injury directly or provoke bystander T cell responses[50]. Administration of αPD-1 or αCTLA-4 has been associated with CMV reactivation in colon, so it is probable that CMV also reactivates in liver during dual therapy[51]. In two patients with high-grade hepatitis, we observed liver inflammation subsided immediately after starting valganciclovir. In our view, the likely presence of viral antigen and apparent responsiveness to valganciclovir is consistent with a direct pathological effect of CMV-specific T cells, although we accept this will be hard to prove in patients.

Better understanding the contribution of anti-viral T cell immunity to checkpoint blockade-related adverse reactions leads us to unexpected solutions for preventing serious complications of cancer immunotherapy. From a clinical perspective, the immunological mechanisms described illustrate how patients' medical histories, especially prior exposure to pathogens, impact their experience of cancer and response to therapy.

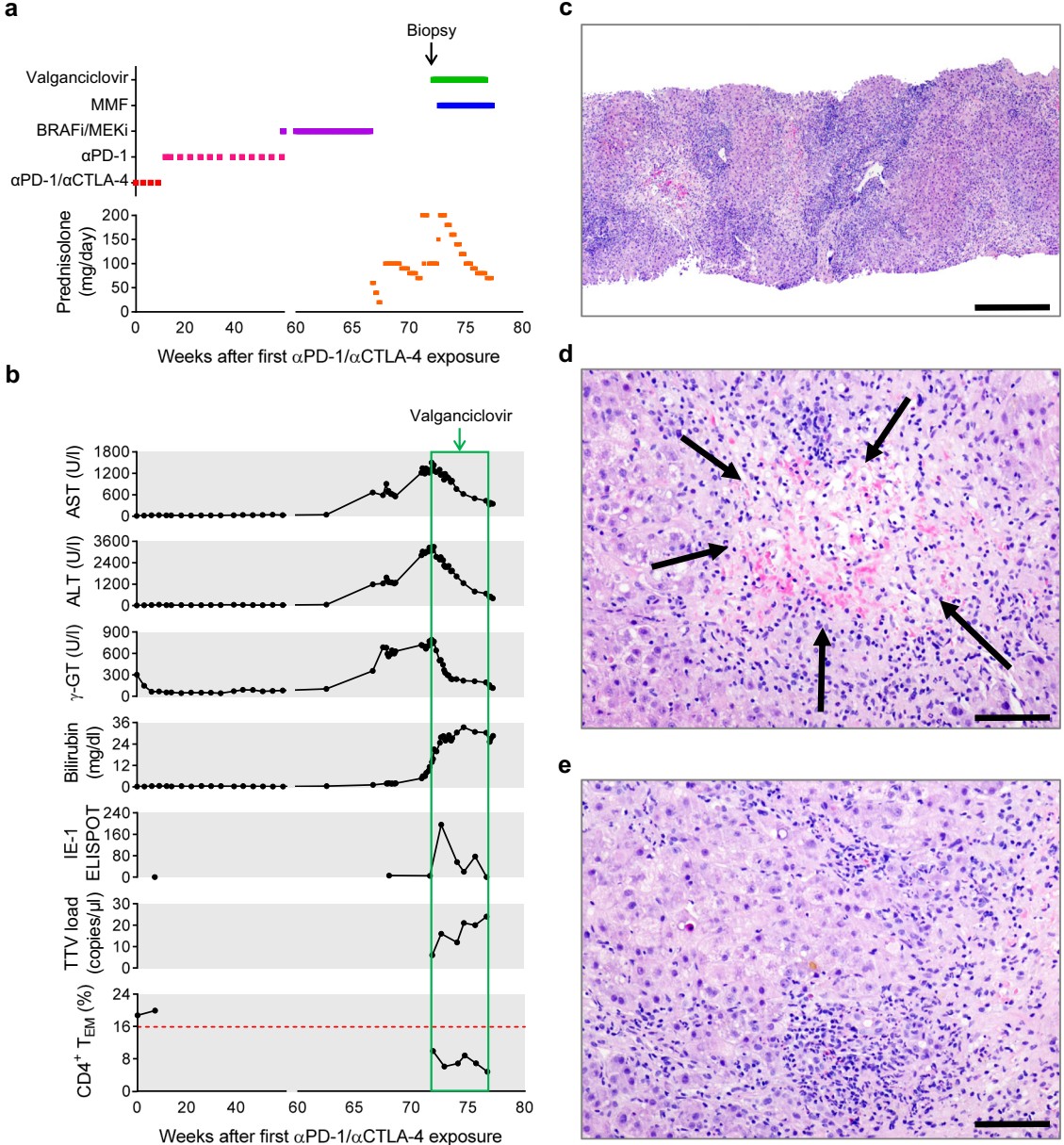

**Fig. 8 Case 2: valganciclovir treatment of late-onset αPD-1/αCTLA-4-related hepatitis.** A 49-year-old male patient who received αPD-1 (Nivolumab) plus αCTLA-4 (Ipilimumab) dual therapy for metastatic melanoma presented with late-onset hepatitis. **a** Course of treatment. **b** Change in hepatitis-related parameters over time and their association with introduction of valganciclovir treatment. **c** Histopathological image of a liver biopsy taken at the start of week 72 showing signs consistent with drug toxicity, autoimmunity or viral infection (H&E staining; scale bar 500 μm). **d** Extensive centrilobular necrosis (arrows) involving 30–40% of hepatocytes was observed (H&E staining; scale bar 100 μm). **e** Dense inflammatory infiltration of lymphocytes, eosinophils and neutrophils was seen in portal areas (H&E staining; scale bar 100 μm).

## Methods

**Study approval and patient management**. This study involving human research participants was performed in accordance with the Declaration of Helsinki and all applicable German and European laws and ethical standards. Specifically, specimens were obtained from patients with Stage III or IV melanoma participating in a single-centre observational clinical trial authorised by the Ethics Committee of the University of Regensburg (approval 16-101-0125) and registered with clinicaltrials. gov (NCT04158544). All participants gave full, informed written consent. The first reported patient was recruited in October 2016 and the last reported patient was recruited in July 2020. Patients received standard-of-care treatment according to local guidelines (Fig. 1a). Stage IV patients with unresectable metastatic disease who received first- or second-line checkpoint inhibitor therapy were initially treated with Nivolumab (αPD-1; Bristol-Myers Squibb) and Ipilimumab (αCTLA-4; Bristol-Myers Squibb) for four cycles, and thereafter with Nivolumab maintenance therapy (3 mg/kg at 3-week intervals). Those patients with complete resection of Stage III melanoma who received adjuvant first-line checkpoint

inhibitor therapy were treated for up to 1 year with Pembrolizumab (αPD-1; MSD) or Nivolumab.

**Clinical flow cytometry**. Detailed step-by-step protocols for preparation and analysis (including specimen gating strategies) of clinical samples by flow cytometry are available through Protocol Exchange[52]. In brief, peripheral blood samples were collected into ethylenediaminetetraacetic acid (EDTA)-vacutainers by venepuncture and then delivered to the immune monitoring laboratory at ambient temperature. Pre-analytical samples were stored for up to 4 h at 4 °C until processing. Whole blood was stained with DuraClone reagents (Duraclone IM Phenotyping Basic Tube, B53309; Duraclone IM T cell Subsets Tube, B53328; Duraclone IM TCRs Tube, B53340; Duraclone IM Treg Tube, B53346; Duraclone IM B cells Tube, B53318; Duraclone IM Dendritic Cells Tube, B53351; all from Beckman Coulter, Krefeld, Germany). Data were recorded with a Navios™ cytometer running Cytometry List Mode Data Acquisition and Analysis Software

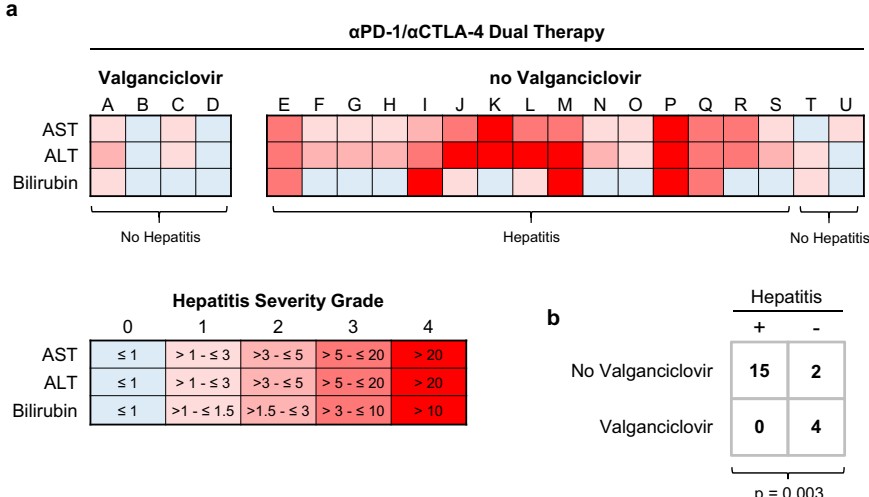

**Fig. 9 Valganciclovir prophylaxis prevents αPD-1/αCTLA-4-related hepatitis. a** Heatmaps showing peak values of AST, ALT and bilirubin expressed as multiples of sex-adjusted upper limit of normal ranges. Fifteen of 17 (88.2%) patients classified as CMV IgG+ CD4+ T_EM≥16% at baseline subsequently developed hepatitis. In contrast, 4 of 4 CMV IgG+ CD4+ T_EM≥16% patients treated with 900 mg/day prophylactic valganciclovir remained hepatitis free. **b** In four initial cases, prophylactic valganciclovir prevented development of hepatitis (n = 21; F.E.).

version 1.3 (Beckman Coulter). Blinded analyses were performed by an experienced operator using Kaluza version 2.1 according to gating strategies illustrated in Supplementary Figs. 2–8.

**Detection of CMV-reactive T cells by flow cytometry**. Detailed step-by-step protocols are available through Protocol Exchange[53]. Briefly, frozen PBMC from patients were thawed and resuspended in RPMI medium (ThermoFisher, Schwerte, Germany) supplemented with 5% heat-inactivated pooled, male-only human AB serum (ZKT Tübingen, Germany), 100 U/ml penicillin, 100 μg/ml streptomycin and 2 mM Glutamax (ThermoFisher). Cells were plated in 96-well round-bottom plates at $1 \times 10^6$ cells in 150 μl/well then incubated at 37 °C with 5% CO₂. After 6 h, cultures were stimulated with 5 μg/ml CMV Lysate (Origene) and αCD28/αCD49 for 18 h. In some conditions, cultures were treated with neutralising monoclonal antibodies against PD-1 (MAB10864, R&D) plus CTLA-4 (MAB3254, R&D) at 5 μg/ml each or mouse IgG1 isotype control (MAB002, R&D) at a final concentration of 10 μg/ml. During the last 4 h of cell culture, protein secretion was blocked with Protein Transport Inhibitor Cocktail (ThermoFisher) containing Brefeldin A und Monensin. Cytokine production was then assessed by flow cytometry. Dead cells were excluded with ViaKrome-808 Fixable Viability Dye (Beckman Coulter). Staining was performed in Cell Staining Buffer (BioLegend) with 10% FcR Block (Miltenyi). Cells were fixed and permeabilised using the eBioscience IC Fixation Buffer and Permeabilization Buffer (eBiosciences, ThermoFisher). Data were recorded with a Cytoflex LX™ cytometer running CytoExpert Software (Beckman Coulter) and analysed using Kaluza version 2.1 (Beckman Coulter) according to the gating strategy shown in Supplementary Fig. 11.

**18F-FDG PET/CT imaging**. Hepatic glucose metabolism as a measure of liver inflammation[34] was analysed retrospectively in all 32 patients who underwent pre-treatment 18F-FDG positron emission tomography and computed tomography (PET-CT) imaging as part of routine tumour staging shortly before starting immunotherapy. After overnight fasting, whole-body PET acquisitions covering at least the trunk were started 60–90 min after i.v. injection of 18F-FDG (3 MBq per kg body weight) on a Biograph Sensation 16 LSO or Biograph mCT40 Flow PET/CT scanner (both Siemens Healthcare, Germany) and lasted 18–24 min depending on patient size. The same area was covered by a low-dose CT scan (tube current <50 mAs, tube voltage 120 kV) if no contrast agents were used. Otherwise 130 ml of Accupaque™ 300 (GE Healthcare) was applied as intravenous contrast agent with consecutive full-dose CT acquisition (120 kV, <100 mAs). PET images (slice thickness 5 mm) were corrected for random coincidences, decay, scatter, and attenuation and reconstructed iteratively using vendor parameter presets. PET and CT images were checked for breathing artifacts and scaled to standardised uptake values (SUV). SUV_mean Blood was measured in a thread-like region-of-interest (ROI) manually set into the aorta and SUV_mean Liver in a spherical ROI in the right liver lobe using the program ROVER (Version v3.0.35, Helmholtz-Zentrum Dresden-Rossendorf, Germany). SUR was calculated as with correction for the distribution time $T$ of 18F-FDG to the reference time $T_0$ of 75 min post injection[54]: $SUR = (T_0/T) \times (SUV_{mean}$ liver / $SUV_{mean}$ blood).

**Routine clinical investigations**. Routine biochemical and haematological investigations were performed by an in-house accredited diagnostic laboratory (Institute of Clinical Chemistry and Laboratory Medicine, UKR). Virological investigations

were conducted by an in-house accredited laboratory (Institute of Clinical Microbiology and Hygiene, UKR) following routine diagnostic procedures. CMV-reactive T cells were quantified by IFN-γ ELISPOT (Lophius, Germany) according to the manufacturer's instructions in an accredited laboratory (UKR).

**Statistics, data transformations and visualisation**. Significance tests and curve-fitting were performed with GraphPad Prism 6.04 (GraphPad Software, Inc, La Jolla, USA) or SPSS® version 25 (IBM Analytics, New York, USA). As indicated, data were analysed using a two-tailed Mann–Whitney (M.W.) test, two-tailed Fisher's exact test (F.E.), D'Agostino & Pearson's normality test ($K2$), one-way or two-way ANOVA, two-tailed t-tests, log-rank survival analysis, receiver-operator characteristic (ROC) analysis or calculating pairwise Pearson correlations ($R^2$). Where stated in the figure legends, p-values were adjusted for multiple comparison using Benjamini–Hochberg (B.H.) correction or Bonferroni correction, but otherwise no adjustment was made.

**Reporting summary**. Further information on research design is available in the Nature Research Reporting Summary linked to this article.

## Data availability
Data supporting the findings of this study are available within the article and its Supplementary information files, or are available from the corresponding author subject to legal restrictions governing use of personal information. Source data are provided with this paper.

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

## Acknowledgements

The authors gratefully acknowledge the financial support of Bristol-Myers Squibb and the BMS Foundation (Award FA-19-009) in the form of unrestricted research grants. This study was partly supported by grants from the German Research Foundation to S.H. (DFG HA 8481/1-1) and J.A.H. (DFG HU 1838/1-1; DFG HU 1838/2-1). L.C. is a Marie Skłodowska-Curie Research Fellow affiliated with INsTRuCT and receives funding from the European Union's Horizon 2020 research and innovation programme (Award 860003). The authors thank Dr. Michael Kapinsky of Beckman Coulter GmbH for advice and support with clinical flow cytometry. The authors also thank Prof. Tobias Pukrop and staff of the Interdisciplinary Center for drug Tumor Therapy (ICT) for supporting collection of clinical samples. This work would not have been possible without the outstanding technical support of Erika Ostermeier, Marina Sorokina, Veronika Menath, Anke Hofmann and Joachim Schweimer.

## Author contributions

J.A.H. designed the immune monitoring studies and other experiments, analysed the results and wrote the manuscript; K.K. organised and performed immune monitoring studies; P.R. designed and performed the experiments; J.J.W., J.K. and B.S. performed the virological analyses; K.E. provided expert histopathological descriptions; M.S., L.B. and D.H. performed the radiological and molecular imaging analyses; G.G., F.Z. and R.S. provided expert statistical and bioinformatics advice; L.C. quality-checked the manuscript; C.B. and R.B. were responsible for the routine clinical laboratory analyses; M.M. and K.D. provided the biological samples; H.L.S. and F.B. performed the experiments; H.J.S. provided infrastructural support; E.K.G., J.M.W. and S.H. provided material support, patient samples and critical feedback throughout the project. All authors approved the final version of the manuscript.

## Funding

## Competing interests

S.H. declares he has received consulting fees and speaker's honoraria from BMS and Merck Sharp & Dohme (MSD). All other authors declare no competing interests.
