## [Peer Review File · Nature Communications]

REVIEWER COMMENTS

Reviewer #1 (Remarks to the Author):

This is a well-designed, prospective, single-center observational study examining mechanisms of the treatment-limiting immune-mediated toxicity hepatitis due to dual checkpoint blockade. It provides hypotheses and insights into possible mechanisms of checkpoint-inhibitor mediated hepatitis, and presents some evidence of a potential predictive biomarker (CD4+TEM^{high}) in those with a history of exposure to CMV, which requires further validation. The authors offer a possible intervention to prevent immune-mediated hepatitis in these patients by prophylactic use of a well-known and widely prescribed anti-viral drug, valganciclovir, although this needs to be tested/trialed properly.

There are some limitations to the data/interpretation that need clarifying; the authors try to link multiple observations that may not be connected at all and some crucial data are based on one patient. Generally, CMV seropositivity is associated with expansion of memory T cells in blood, and with age it goes up. It is possible that most of the data could be explained by this.

Major comments:

1. There is a lack of data showing CMV reactivation in the liver.
2. Are the expanded effector memory CD4+ T cells in metastatic melanoma patients CMV-reactive? This needs to be shown. In figure 5d, there are more CMV-reactive T cells in CD4 TEM^{hi} patients. Firstly, it is only 1 patient, although it is claimed that it is representative (how many patients were tested? Please elaborate). Secondly, increased CMV-reactive CD4s in these patients is not surprising. It is well-known that CMV seropositive patients generally have a higher proportion of effector memory T cells in blood (as acknowledged by the authors in the intro). So, if the groups are split on the basis of CD4+ TEM numbers, naturally patients who are CMV+ will have higher levels. What is figure 5c demonstrating? Please elaborate and clarify.
3. Why do only some CMV+ patients develop hepatitis on checkpoint blockade therapy? Why does CMV reactivate only in some metastatic patients? Associated with baseline burden of melanoma?
4. Although CMV-induced hepatitis has been observed in highly immunocompromised patients (often in patients who lack total T cells), the authors do not provide data to support the claim that metastatic tumours compromise CMV immunity. The only link is increased CD4+ TEM (bit of a chicken and the egg question here).
5. Please provide an explanation for how the liver is damaged. Viral reactivation or T cell responses to the virus? Guessing it is the later (otherwise it will be independent of checkpoint blockade), but it is not clear if CMV infects hepatocytes. It is also not clear where the damage is in the tox patients. The authors discuss potential bystander activation, but that is handwaving. Please clarify and elaborate on these issues.

Minor comments:

1. Hepatic F-FDG uptake data is an indirect evidence for liver inflammation. They don't talk about any other markers, including CRP.
2. The only patient who responded to valganciclovir after checkpoint blockade-induced hepatitis had below the cut off level of CD4+TEM
3. Prophylactic valganciclovir treatment was only done on one patient and how do we know this patient would have certainly got hepatitis without it?

4. Sample numbers are too low to call this a predictive marker
5. The authors claim that the expansion of CD4 TEM was only seen in patients with metastatic tumours and not in patients with fully resected primary tumour – please provide the ages of these two groups. Is it possible, those with metastatic tumours happened to be older?
6. Abstract. The mention of exposure of pathogen (CMV in this case) shaping clinical responses (generally meaning tumor response) to cancer therapy is confusing and should be clarified.
7. Introduction: Paragraph 2, while mentioning current aetiological understandings of toxicities, you generalize that T cell dysregulation drives these although autoantibody amplification and complement activation are important pathways contributing to immune-toxicity (Young A et al, The Balancing act between cancer immunity and autoimmunity in response to immunotherapy, Clinical immunology research, 2018) and can be clarified here.
8. Introduction: Paragraph 4, you may mention evidence for the “memory inflation” phenomena (as described in CMV-specific CD8+ T cell) in CD4+ T cells, if any. Otherwise, suggesting similar phenomenon in CMV-specific CD4+ T cells should be clarified as this is beyond the aim and scope of this study.
9. Introduction: Paragraph 5, it can be “hypothesized” that reactivation of CMV (or CMV-specific CD+T cells) is contributing to pathogenesis in hepatitis (as no immunohistochemical evidence provided in this study to exclude CMV-mediated immune hepatitis).
10. Results: Fig 1b- It is important to show the incidence of organ-specific grade (1-4) toxicity in this cohort (instead of the number of complications) in keeping with standard toxicity reporting guideline (ie CTCAE) which allows comparison of cohort’s incidence to other patients (clinical trials or real-world data).
11. Results: Suppl 3, 4- Can it be clarified that all responses are evaluated as per Response Evaluation Criteria in Solid Tumor (RECIST)? For example, “mixed response” is rather a clinical term, not specific to RECIST.
12. Results: Fig 2h/l, 2 patients with CD4+TEMhigh that were prospectively treated with anti PD1 (not included in study cohort), the CMV status and other biochemical parameters can be shown/mentioned to minimize bias.
13. Results: Fig 3e-f and fig 4f-n, expansion of effector compartment in CD+ and CD8+ is shown along without expansion of exhausted T cells (ie PD1+), was CTLA4+ CD4/CD8+ analyzed/included? Would be interesting to show as the results are exclusive to dual checkpoint blockade or more precisely, CTLA 4 blockade.
14. Results: Fig 6, SURmean liver was lowest in CD4+TEMhigh indicating least inflamed liver compared with other groups; important to show the percentage of CMV+ in this group.
15. Results: Fig 7, standard combination immunotherapy is given for 4 cycles though the patient received 3 cycles. You may explain the reason for elevated bilirubin on the recurrence of hepatitis.
16. Discussion: The discussion could be stronger and more constructive highlighting the importance of a potential biomarker, CD4+TEMhigh in immune-hepatitis while mentioning other biomarker studies in this field. While providing hypothesis regarding mechanistic of CD4+ mediated tissue injury, you may quote reference for pattern of tissue injury in immune-toxicity.
17. It would be interesting to know the on-treatment changes of CD4+TEMhigh and other T cell compartment in this cohort or a subset of pts.

Reviewer #2 (Remarks to the Author):

This is an interesting paper that addresses the potential role of CMV infection as risk factor for hepatitis in checkpoint blockade.

The concern relates to subset analysis in a relatively modest cohort, although the difficulty in acquiring such a cohort in the real world is recognised.

More discussion of the role of T cell migration between tissue and blood would be helpful. When a T cell is rare in blood it usually means that it has gone into a tissue. How do these findings relate to infiltration of cells into liver? Why are naive cells lower in this group?

- I wonder if the association with CD4EM is overstated. "In the training set, our model had a positive predictive value (PPV) of 100% and sensitivity of 42 %. Test validation was performed in 30 ..patients: 4 of 4 patients with CD4+ TEM \geq 21.1 % developed hepatitis".

What is the sensitivity and specificity in the test cohort ?

The combined cohort data will look good as it includes the large training cohort.

- Is CMV serostatus alone a risk factor for hepatitis or any other auto-immune side effect ?

-The CD4TEM counts are not different. The percentage is altered due to a decrease in naive cells.

The memory response is the same - but the text says this suggests there is chronic antigen stimulation?

- I note the stimulation of CD4+ T cells with CMV peptides in vitro. (CMV viral lysate is a better assay for this). Was there any difference between CMV+ patients who did or did not develop hepatitis?

-the seasonal association is interesting. Is this seen in large cohorts of checkpoint therapy ?

- There is less inflammation in the CD4TEMhigh group. The etiological sequence on page 10 suggests a mechanism, but CD4EM cells are not expanded in this group ? (naive numbers are down)

- The case reports are interesting. The treatment case appears to be in a low risk CD4TEMlow group. Prophylaxis was done in one case.

The data are interesting and provocative, but there are some concerns at this stage about statistical validity.

Also somewhat overstated as a mechanism.

Reply to Reviewer 1

Remarks to the Author: This is a well-designed, prospective, single-center observational study examining mechanisms of the treatment-limiting immune-mediated toxicity hepatitis due to dual checkpoint blockade. It provides hypotheses and insights into possible mechanisms of checkpoint-inhibitor mediated hepatitis, and presents some evidence of a potential predictive biomarker (CD4+TEM^{high}) in those with a history of exposure to CMV, which requires further validation. The authors offer a possible intervention to prevent immune-mediated hepatitis in these patients by prophylactic use of a well-known and widely prescribed anti-viral drug, valganciclovir, although this needs to be tested/trialed properly.

There are some limitations to the data/interpretation that need clarifying; the authors try to link multiple observations that may not be connected at all and some crucial data are based on one patient. Generally, CMV seropositivity is associated with expansion of memory T cells in blood, and with age it goes up. It is possible that most of the data could be explained by this.

Thank you for your insightful and detailed critique. Your comments forced us to distinguish more clearly between *most likely* and *proven* mechanisms, and to concede there are some questions that cannot be answered definitively in humans.

“Normal” age-related expansion of CMV-reactive memory T cells cannot explain the level of pre-therapy expansion of CD4⁺ T_{EM} cells in at-risk patients, even allowing for CMV status and age. The median age (IQR) in our at-risk patients was 68 (58 – 77) and their median CD4⁺ T_{EM} % (IQR) was 28.3 (26.1 – 33.3). We previously reported the normal range for CD4⁺ T_{EM} % (IQR) in 60 – 75 year olds as 15.5 (12.2 – 23.8) [Kverneland AH, Cytometry A, 2016]. Our comparison between patients with fully resected and metastatic disease on p.7 and Figure 2e underscores this point. Please also refer to our answers to Q2b and Q10.

(1) There is a lack of data showing CMV reactivation in the liver.

This is correct and presents a challenge to our theory. However, it's not that we have negative findings, but that we are unable to test for CMV reactivation in liver. Biopsy is an invasive procedure with risk of serious complications. There is no clinical justification for liver biopsy at baseline or at first presentation of hepatitis in most patients.

We were able to investigate biopsies from two patients who developed severe hepatitis late after treatment. These cases are now described on p.11-13 and figures 7&8. Although we were unable to detect replicating virus, this is not entirely surprising in a generally

immunocompetent patient. It is conceivable the virus that stimulated liver inflammation was gone by the time a biopsy was performed. Apart from control of viral reactivation by T cell immunity, it is possible that viral replication capable of sustaining a T cell response is simply below our detection limit.

We argue the effect of valganciclovir in our patients demonstrates that replicating CMV is necessary for developing hepatitis in these cases. In light of the immunological evidence of CMV responses in at-risk patients, we should not over-interpret the lack of direct evidence of CMV replication.

(2) Are the expanded effector memory CD4⁺ T cells in metastatic melanoma patients CMV-reactive? This needs to be shown. **(a)** In figure 5d, there are more CMV-reactive T cells in CD4⁺ TEM^{hi} patients. Firstly, it is only 1 patient, although it is claimed that it is representative (how many patients were tested? Please elaborate). Secondly, increased CMV-reactive CD4s in these patients is not surprising. It is well-known that CMV seropositive patients generally have a higher proportion of effector memory T cells in blood (as acknowledged by the authors in the intro). **(b)** So, if the groups are split on the basis of CD4⁺ TEM numbers, naturally patients who are CMV⁺ will have higher levels. What is figure 5c demonstrating? Please elaborate and clarify.

This entire set of experiments was repeated using CMV lysates to stimulate patient PBMC. Please refer to the new section on p.11, new Figure 6, supplementary Figure 10 and the new supplementary technical details. **(a)** We now have quantitative data from n=44 patients. We do not dispute that CMV-reactive T cells accumulate as people grow older; however, the level of CD4⁺ T_{EM} cell expansion in at-risk patients is beyond normal, even allowing for age [Kverneland AH, Cytometry A, 2016]. This is emphasised by the comparison between patients with metastatic and fully resected disease (Figure 2d). **(b)** The proportion of CMV-reactive CD4⁺ T cells in CMV IgG⁺ CD4⁺ T_{EM}^{high} patients is higher than CMV IgG⁺ CD4⁺ T_{EM}^{low} patients:

- (3) (a)** Why do only some CMV+ patients develop hepatitis on checkpoint blockade therapy?
(b) Why does CMV reactivate only in some metastatic patients? Associated with baseline burden of melanoma?

This is an important question, which we now address directly on pg.10-11 and Figure 5f. **(a)** Within the scope of this article, our answer is, because only some CMV $^+$ patients experience baseline expansion of CD4 $^+$ T $_{EM}$ cells. This could be because (i) CMV only reactivates in a subset of patients, (ii) CMV reactivates but conditions for expansion of CD4 $^+$ T $_{EM}$ cells only occur in some individuals, or (iii) a third factor (such as coinfection with a winter virus) is necessary. **(b)** We do not claim that CMV reactivates only in some patients – it may well reactivate in all patients, we have no evidence either way – but only a subset of patients undergoes CD4 $^+$ T $_{EM}$ expansion. None of the clinical or biochemical features we examined in Figure 3 or Suppl. 8 explains this difference, including indices of tumour burden.

- (4)** Although CMV-induced hepatitis has been observed in highly immunocompromised patients (often in patients who lack total T cells), the authors do not provide data to support the claim that metastatic tumours compromise CMV immunity. The only link is increased CD4 $^+$ TEM (bit of a chicken and the egg question here).

We concede this point and have softened our interpretation throughout the article.

- (5)** Please provide an explanation for how the liver is damaged. Viral reactivation or T cell responses to the virus? Guessing it is the later (otherwise it will be independent of checkpoint blockade), but it is not clear if CMV infects hepatocytes. It is also not clear where the damage is in the tox patients. The authors discuss potential bystander activation, but that is handwaving. Please clarify and elaborate on these issues.

Please refer to page 13 and Figure 9. Here, we show that 4 of 4 CMV IgG $^+$ CD4 $^+$ T $_{EM}^{high}$ patients given prophylactic valganciclovir remained hepatitis-free, whereas 15 of 17 patients

who did not receive valganciclovir developed hepatitis. We are unable to detect the virus, so we are quite sure that liver damage is an immune-mediated pathology, not a cytopathic effect of CMV. Please refer to our modified discussion on p.15.

(6) Hepatic F-FDG uptake data is an indirect evidence for liver inflammation. They don't talk about any other markers, including CRP.

Thanks, we have corrected this deficiency. Please refer to Figure 3 and Suppl. 2, 5 and 8.

(7) The only patient who responded to valganciclovir after checkpoint blockade-induced hepatitis had below the cut off level of CD4+TEM

Please refer to p.10 for our revised explanation of cut-offs, as well as the new case reports and prophylactically treated patients in figures 7, 8 and 9.

(8) Prophylactic valganciclovir treatment was only done on one patient and how do we know this patient would have certainly got hepatitis without it?

Please refer to Figure 9. We now report n=4 treated versus 17 untreated. $p = 0.003$

(9) Sample numbers are too low to call this a predictive marker

As directed by the editor, we added more patients. CD4⁺ T_{EM} % in CMV IgG⁺ patients is a good discriminator for developing hepatitis (AUROC = 0.89). Our article has been reviewed by 3 professional statisticians, including the Professor of Statistical Bioinformatics, who are all coauthors (GG, FZ and RS). All claims made in our article are statistically valid.

(10) The authors claim that the expansion of CD4 TEM was only seen in patients with metastatic tumours and not in patients with fully resected primary tumour – please provide the ages of these two groups. Is it possible, those with metastatic tumours happened to be older?

CD4⁺ T_{EM} % does not vary with age in our cohort or other published human data [Kverneland AH, Cytometry A, 2016]. The mean age [95% CI] of fully resected patients was 63.3 [60.4 – 68.3] years, whereas the mean age [95% CI] of metastatic patients was 59.7 [56.9 – 62.5] years ($p = 0.06$; t-test, 2-tailed, equal var).

(11) Abstract. The mention of exposure of pathogen (CMV in this case) shaping clinical responses (generally meaning tumor response) to cancer therapy is confusing and should be clarified.

Thank you, we now use “reactions after” instead of “responses to”

(12) Introduction: Paragraph 2, while mentioning current aetiological understandings of toxicities, you generalize that T cell dysregulation drives these although autoantibody amplification and complement activation are important pathways contributing to immunotoxicity (Young A et al, The Balancing act between cancer immunity and autoimmunity in response to immunotherapy, Clinical immunology research, 2018) and can be clarified here.

Thank you, this change and citation are made on p.4.

(13) Introduction: Paragraph 4, you may mention evidence for the “memory inflation” phenomena (as described in CMV-specific CD8+ T cell) in CD4+ T cells, if any. Otherwise, suggesting similar phenomenon in CMV-specific CD4+ T cells should be clarified as this is beyond the aim and scope of this study.

Thank you. For simplicity, we’ve dropped this interpretation.

(14) Introduction: Paragraph 5, it can be “hypothesized” that reactivation of CMV (or CMV-specific CD+T cells) is contributing to pathogenesis in hepatitis (as no immunohistochemical evidence provided in this study to exclude CMV-mediated immune hepatitis).

We’ve changed “pathogenesis” to “immunopathogenesis” on p.5 to avoid misunderstanding.

(15) Results: Fig 1b- It is important to show the incidence of organ-specific grade (1-4) toxicity in this cohort (instead of the number of complications) in keeping with standard toxicity reporting guideline (ie CTCAE) which allows comparison of cohort’s incidence to other patients (clinical trials or real-world data).

Our point relates to the coincidence of complications. If they were tightly associated, it might suggest a common aetiology. We report CTCAE grading elsewhere (Figure 2h).

(16) Results: Suppl 3, 4- Can it be clarified that all responses are evaluated as per Response Evaluation Criteria in Solid Tumor (RECIST)? For example, “mixed response” is rather a clinical term, not specific to RECIST.

Our initial objective was to identify predictive markers that are relevant to everyday dermatological practice. RECIST score is not routinely calculated for our patients. Our point in showing this data is related to Q(15) – incidence of hepatitis is not closely associated with clinical outcome; therefore, we are justified in looking for alternative immunological mechanisms. For this manuscript, we do not think the overwhelming amount of work to assess RECIST scores is justified by the additional accuracy it might bring.

(17) Results: Fig 2h/l, 2 patients with CD4+TEMhigh that were prospectively treated with anti PD1 (not included in study cohort), the CMV status and other biochemical parameters can be shown/mentioned to minimize bias.

In the current version, these patients are discussed before involvement of CMV is introduced (Figure 2e & p7). All 4 patients were CMV IgG⁺.

(18) Results: Fig 3e-f and fig 4f-n, expansion of effector compartment in CD4+ and CD8+ is shown along without expansion of exhausted T cells (ie PD1+), was CTLA4+ CD4/CD8+ analyzed/included? Would be interesting to show as the results are exclusive to dual checkpoint blockade or more precisely, CTLA 4 blockade.

This is a very interesting question, especially given the results from our monotherapy patients, who did not receive αCTLA-4 therapy and did not develop hepatitis. Regrettably, we didn't measure CTLA-4 expression with any of our flow cytometry panels.

(19) Results: Fig 6, SURmean liver was lowest in CD4+TEMhigh indicating least inflamed liver compared with other groups; important to show the percentage of CMV+ in this group.

We've changed the order of the manuscript, so this question no longer applies; however, all CD4⁺ T_{EM}^{high} patients were CMV IgG⁺.

(20) Results: Fig 7, standard combination immunotherapy is given for 4 cycles though the patient received 3 cycles. You may explain the reason for elevated bilirubin on the recurrence of hepatitis.

The patient experienced joint pain, so the fourth cycle was cancelled on the planned day of infusion. This information is now included in the case description on p.11-12 and Figure 7.

(21) Discussion: The discussion could be stronger and more constructive highlighting the importance of a potential biomarker, CD4+TEMhigh in immune-hepatitis while mentioning other biomarker studies in this field. While providing hypothesis regarding mechanistic of CD4+ mediated tissue injury, you may quote reference for pattern of tissue injury in immunotoxicity.

We see this as a mechanistic description rather than a biomarker study. It concerns a very specific subset of patients and possible treatment options for them. Although there's a lot of important work on biomarkers in checkpoint blockade, we feel our Discussion is not the right place to review it, especially given the current length of our article.

(22) It would be interesting to know the on-treatment changes of CD4+TEMhigh and other T cell compartment in this cohort or a subset of pts.

This supplementary figure was removed from an earlier version of our manuscript after editorial review because it was too peripheral to our story:

Supplementary Figure 1: Stability of CD4+ TEM^{high} phenotype. Patients who were CD4+ TEM^{high} at baseline maintained their phenotype after αPD-1/αCTLA-4 treatment. Strong induction of CD4+ TEM cell responses at 3 or 6 weeks were observed in many patients who were CD4+ TEM^{low} at baseline, but this did not predict hepatitis.

Reply to Reviewer 2

This is an interesting paper that addresses the potential role of CMV infection as risk factor for hepatitis in checkpoint blockade.

Thank you for your assessment and challenging questions, particularly for your suggestion to repeat our stimulation experiments with CMV lysates, which worked nicely.

(I) The concern relates to subset analysis in a relatively modest cohort, although the difficulty in acquiring such a cohort in the real world is recognised.

Since you last saw this manuscript, we added another 19 patients to our model-building sets. We now have n=4 CD4+ TEM^{high} patients with metastatic disease who were treated with αPD-1 monotherapy. Furthermore, we now have 2 patients with hepatitis who were treated with valganciclovir (Figure 7&8) and another 4 patients who received prophylactic valganciclovir (Figure 9). Our manuscript has been critically reviewed by 3 professional statisticians. We

acknowledge that n=103 is a modest number of patients; however, all claims made in our manuscript are statistically valid.

(II) (a) More discussion of the role of T cell migration between tissue and blood would be helpful. When a T cell is rare in blood it usually means that it has gone into a tissue. How do these findings relate to infiltration of cells into liver? (b) Why are naive cells lower in this group?

This is a fascinating point, but we removed this aspect of the manuscript after an initial editorial review. (b) We think the peripheral CD4+ T cell pool is more-or-less constant, but that chronic or recurrent low-level restimulation of CMV-reactive T cells preferentially drives T_{EM} over T_{CM} differentiation. These data are from Figure 4:

(1) I wonder if the association with CD4EM is overstated. "In the training set, our model had a positive predictive value (PPV) of 100% and sensitivity of 42 %. Test validation was performed in 30 patients: 4 of 4 patients with CD4+ TEM ≥ 21.1 % developed hepatitis". What is the sensitivity and specificity in the test cohort? The combined cohort data will look good as it includes the large training cohort.

Your points are well-taken. In this revision, we were stricter about keeping our training and validation sets separated, especially in Figure 2 and Suppl. 10. We provide measures of model performance just for the validation set. Please refer changes on p.6-7 and p.10.

(2) (a) Is CMV serostatus alone a risk factor for hepatitis or any other auto-immune side effect? (b) The CD4TEM counts are not different. The percentage is altered due to a decrease in naive cells. The memory response is the same - but the text says this suggests there is chronic antigen stimulation?

(a) We regard CMV antibodies as a marker of infected individuals, not as a direct cause of hepatitis. In our model, reactivation of latent CMV drives an atypical T cell response in patients with metastatic disease. Together, CMV serostatus and T_{EM} expansion are highly

discriminatory (AUROC = 0.885) for patients at risk of hepatitis. However, CMV serostatus alone is not a useful predictive marker for hepatitis, colitis, any other clinical complications, or clinical response to therapy. Please refer to our new explanation of cut-offs on page 10 and Suppl. 10. (b) Please refer to p8/9, Figure 4 a-d and Suppl. 8. We think the size of the CD4⁺ T cell pool in blood is constant. In absolute terms, naïve T cell and T_{CM} cell counts are lower in CD4⁺ T_{EM}^{high} patients, whereas CD4⁺ T_{EM} and CD4⁺ T_{EMRA} counts are higher. Overall, this picture is consistent with descriptions from animal models that suggest chronic, low-level stimulation of T cells of preferential drives differentiation of T_{EM} cells (refs. 43-49).

(3) I note the stimulation of CD4⁺ T cells with CMV peptides in vitro. (CMV viral lysate is a better assay for this). Was there any difference between CMV⁺ patients who did or did not develop hepatitis?

We ran our experiments with CMV lysates using PBMC from n=44 patients. Please refer to Figure 6 and corresponding text on p.11. We found a higher proportion of CMV-reactive CD4⁺ T cells in CMV IgG⁺ CD4⁺ T_{EM}^{high} patients than CMV IgG⁺ CD4⁺ T_{EM}^{low} patients. As we would predict, this difference was partly explained by a higher proportion of CMV-reactive CD4⁺ T cells within the CD4⁺ T_{EM} compartment. These data are from Figure 6:

(4) The seasonal association is interesting. Is this seem in large cohorts of checkpoint therapy?

The seasonal influence was seen in cases of CD4⁺ T_{EM}^{high} patients with hepatitis treated with combined αPD-1/αCTLA-4 therapy. The effect is probably too dilute to detect in unstratified studies without immune monitoring results. To the best of our knowledge, this is the first report of such an effect.

(5) There is less inflammation in the CD4⁺ T_{EM}^{high} group. The etiological sequence on page 10 suggests a mechanism, but CD4⁺ T_{EM} cells are not expanded in this group? (naive numbers are down)

Please refer to our answer to Q2a about CD4⁺ T_{EM} cells expansion. Our proposed aetiological sequence is that CD4⁺ T_{EM} cells expand to control a low-level chronic or recurrent reactivation of CMV. This response is successful in controlling CMV, in so far as patients do not develop overt CMV disease. Therefore, we do not expect liver inflammation in these patients at baseline. The problem arises when, after αPD-1/αCTLA-4 therapy is given, CMV reactivates in the liver and causes an exaggerated T cell reaction.

(6) The case reports are interesting. The treatment case appears to be in a low risk CD4TEMlow group. Prophylaxis was done in one case.

Please refer to p.10 and Suppl.10 for our revised explanation of cut-offs. Please also refer to the new case of valganciclovir treatment of hepatitis, as well as the newly added prophylactically treated patients in figures 7, 8 and 9.

With our new patients, we now give you significant p-values:

		Hepatitis				Hepatitis	
		+	-			+	-
DT: αPD1 + αCTLA4		12	0	No Valganciclovir		15	2
MT: αPD1		1	3		Valganciclovir		0
		p = 0.007				p = 0.003	

(7) The data are interesting and provocative, but there are some concerns at this stage about statistical validity. Also somewhat overstated as a mechanism.

What we present is statistically valid, despite the modest sample size of n=103 recruited dual therapy patients. The most direct proof of our proposed mechanism we can give in humans is the effectiveness of valganciclovir. In the limited number of patients (n=6) we've treated so far, valganciclovir seems to have a positive effect on markers of liver injury. For this reason, we see this article as the basic immunological prelude to a larger prospective RCT of valganciclovir prophylaxis. If you are persuaded such a trial would be worthwhile, then this article is a success.

REVIEWERS' COMMENTS

Reviewer #1 (Remarks to the Author):

The authors have provided some clarifications and the revised manuscript with additional data adds strength to the study. It is clear from the data presented in this manuscript that increased CD4+ TEM levels in the blood is predictive of hepatitis following combination therapy. In this regard the addition of the validation cohort was essential.

However, where I am still not convinced is the link between increased CD4+ TEM, CMV infection and hepatitis. From the diagram provided in the rebuttal letter, the authors claim that subclinical CMV reactivation precedes the expansion of CD4+ TEM. Unfortunately, there is no evidence provided for this claim. The only supportive data is CMV seropositive patients who subsequently developed hepatitis had higher levels of CD4TEM. It is not clear whether the expanded TEM was CMV-specific. The additional data provided with in vitro stimulation is not convincing. There is only a marginal difference in CMV-reactive T cells between CMV IgG+ CD4 TEM <15.9% and CMV IgG+ CD4TEM >15.9% (possible difference of ~0.2%). The difference between CMV seropositive response and seronegative response is not convincing either. One would expect ~4% of CD4+ T cells and ~9% of memory CD4+ T cells to be CMV reactive (doi: 10.1084/jem.20050882).

The second claim that is not convincing is the potential reactivation of CMV in the liver. Usually, CMV associated tissue disease is diagnosed on the basis of IHC staining for viral antigens (immediate early proteins of CMV) and/or histopathological finding of viral inclusion bodies/cytopathic effect. The authors state that no evidence of viral reactivation was found in two liver biopsy specimens tested. The authors argue that this could be due to low viral replication or possible viral clearance by T cells. While that could be the case, to make the argument that there is CMV reactivation in the liver, we need some convincing evidence. Treatment of 2 patients with Valganciclovir did improve those patients, however, that was in combination with prednisolone. Although the first patient did remain stable even after the withdrawal of prednisolone, it is not enough to convince that valganciclovir alone made the difference. Another line of evidence presented is the prophylactic treatment with valganciclovir. Again, with only n=4, it is hard to link this to a possible CMV reactivation in the liver.

The data presented is intriguing and warrants more studies. However, the current form of the manuscript overstates the data. At best, the authors could claim that CMV IgG together with high CD4TEM is predictive and there is a potential association between CMV and hepatitis. The causal relationship with CMV infection is not accurate. There is no evidence in the literature to suggest CMV reactivation is seasonal either.

Suggestions

1. The authors should tone down the claims – especially the title and abstract. The data presented does not show the expanded TEM population is virus-specific. The data presented also do not support CMV reactivation in the liver. It is appropriate to call it an association rather than a causal effect.
2. Or provide additional data – some class II CMV tetramers are currently available. Perhaps staining CD4+ TEM population with tetramers may show a better enrichment? In vitro stimulation of sorted TEM CD4+ T cells may also provide greater response. If longitudinal blood samples are available, a spike in CMV viral load may suggest reactivation.

Reviewer #2 (Remarks to the Author):

Significant improvements have been made to the manuscript.

The authors have addressed the points to the best of their ability and the cohort size of 101 is strong in an area of unmet medical need.

Reply to Reviewer 1

The authors have provided some clarifications and the revised manuscript with additional data adds strength to the study. It is clear from the data presented in this manuscript that increased CD4⁺ TEM levels in the blood is predictive of hepatitis following combination therapy. In this regard the addition of the validation cohort was essential.

However, where I am still not convinced is the link between increased CD4⁺ TEM, CMV infection and hepatitis. From the diagram provided in the rebuttal letter, the authors claim that subclinical CMV reactivation proceeds the expansion of CD4⁺ TEM. Unfortunately, there is no evidence provided for this claim. The only supportive data is CMV seropositive patients who subsequently developed hepatitis had higher levels of CD4⁺TEM. It is not clear whether the expanded TEM was CMV-specific.

We are grateful to the reviewer for his/her critical appraisal of our work.

We do not claim in the previous or current version of the manuscript to have shown CMV reactivation before baseline. It is entirely reasonable to offer our preferred explanation of the results in our Discussion provided we present it as such. We modified the text on pages 14 and 15 to emphasise the distinction between what was shown and what we consider as the most likely interpretation.

It should be pointed out that testing whether compartmentalised CMV reactivation occurs in melanoma patients before treatment would be very difficult. Firstly, we don't know when or for how long virus might be detectable. There is no clinical or ethical justification for delaying the treatment of such patients in order to collect serial samples for virological investigation. Secondly, if CMV reactivation occurs at a low level within tissues, we would have to perform biopsies to prove it. There is no clinical or ethical justification for this invasive procedure.

It is well-described in animal models that chronic, low-level exposure to antigens preferentially drives effector memory responses (ref. 43-49). In our patients, baseline CD4⁺ T_{EM} expansion is a feature of CMV-infected individuals with generally higher anti-CMV IgG levels and a higher proportion of CMV-reactive CD4⁺ T cells. Our study establishes that CD4⁺ T_{EM} expansion predicts treatment-related hepatitis and that Valganciclovir modifies this risk. The simplest, and therefore preferred, synthesis is a CMV-driven T cell response.

The additional data provided with in vitro stimulation is not convincing. There is only a marginal difference in CMV-reactive T cells between CMV IgG⁺ CD4⁺ TEM <15.9% and CMV IgG⁺ CD4⁺TEM >15.9% (possible difference of ~0.2%). The difference between CMV

seropositive response and seronegative response is not convincing either. One would expect ~4% of CD4⁺ T cells and ~9% of memory CD4⁺ T cells to be CMV reactive (doi: 10.1084/jem.20050882).

Our results show that patients with baseline CD4⁺ T_{EM} expansion have a significantly higher proportion of CMV-responsive CD4⁺ T cells in their repertoire than either uninfected or CMV-infected CD4⁺ T_{EM}^{low} patients. We do not claim our assay method detects all CMV-reactive T cells, but this is not necessary for our conclusion. The reviewer compares our results to the frequencies of CD4⁺ T cells reported by Sylwester et al. (2005) but this is unfair. Sylwester summed the proportion of reactive T cells from multiple cultures, each stimulated with different large peptide pools. Following the recommendation of Reviewer 2, we stimulated our PBMC with CMV lysates. By comparison to our method, Sylwester must've been using much greater amounts of antigen. We also note Sylwester studied healthy human subjects, not patients with advanced melanoma.

The second claim that is not convincing is the potential reactivation of CMV in the liver. Usually, CMV associated tissue disease is diagnosed on the basis of IHC staining for viral antigens (immediate early proteins of CMV) and/or histopathological finding of viral inclusion bodies/cytopathic effect. The authors state that no evidence of viral reactivation was found in two liver biopsy specimens tested. The authors argue that this could be due to low viral replication or possible viral clearance by T cells. While that could be the case, to make the argument that there is CMV reactivation in the liver, we need some convincing evidence.

In the previous and current versions of our manuscript, we provide reasons why replicating virus may have been undetectable in these two cases at the time of biopsy. Obtaining liver biopsies from patients receiving checkpoint blockade can only be justified in the case of steroid-refractory hepatitis. Therefore, we argue the only way of testing our hypothesis that CMV plays a causal role in checkpoint blockade-related is treating with Valganciclovir.

Treatment of 2 patients with Valganciclovir did improve those patients, however, that was in combination with prednisolone. Although the first patient did remain stable even after the withdrawal of prednisolone, it is not enough to convince that valganciclovir alone made the difference.

We do not claim that Valganciclovir treatment alone was responsible for the clinical improvement. By the same token, the treatment responses also cannot be attributed to prednisolone alone. If the reviewer accepts that treatment with Valganciclovir improved the patients, then the conclusion that CMV plays a causal role in checkpoint blockade-related hepatitis is inescapable.

Another line of evidence presented is the prophylactic treatment with valganciclovir. Again, with only n=4, it is hard to link this to a possible CMV reactivation in the liver.

Although we should be cautious about interpreting these four cases, the reviewer cannot simply dismiss a significant result ($p=0.003$; post-hoc power > 0.95). The reviewer will surely agree the appropriate next step is a randomised, prospective, blinded, (ideally) multicentre clinical trial designed to test the efficacy of Valganciclovir or Letermovir prophylaxis in preventing hepatitis in patients with baseline CD4⁺ T_{EM} expansion.

The data presented is intriguing and warrants more studies. However, the current form of the manuscript overstates the data. At best, the authors could claim that CMV IgG together with high CD4TEM is predictive and there is a potential association between CMV and hepatitis. The causal relationship with CMV infection is not accurate. There is no evidence in the literature to suggest CMV reactivation is seasonal either.

Suggestion 1: The authors should tone down the claims – especially the title and abstract. The data presented does not show the expanded TEM population is virus-specific. The data presented also do not support CMV reactivation in the liver. It is appropriate to call it an association rather than a causal effect.

We have moderated our claims throughout the manuscript and properly qualified our opinions as such. These changes are marked in track-change mode on p.11, p.14 and p.15.

Suggestion 2. Or provide additional data – some class II CMV tetramers are currently available. Perhaps staining CD4+ TEM population with tetramers may show a better enrichment? In vitro stimulation of sorted TEM CD4+ T cells may also provide greater response.

We tried using Class II tetramers, but each tetramer only detects T cells specific for a single antigen and tetramers must be HLA-matched with the patient. TCR repertoire profiling may finally resolve this question, but is beyond the scope of our current manuscript.

If longitudinal blood samples are available, a spike in CMV viral load may suggest reactivation.

This is also something we've tried with samples collected after baseline. Of 95 baseline serum samples, 1 patient was positive by CMV-PCR. We also detected a CMV+ saliva sample from a patient at Week 3, in addition to a CMV+ faeces sample from an asymptomatic patient after treatment. Although we regard these findings in 3 patients as abnormal, we do not have a suitable control group to prove it.